# Improved Robust Estimation for Erdős-Rényi Graphs: The Sparse Regime and Optimal Breakdown Point

**Hongjie Chen**
ETH Zürich
hongjie.chen@inf.ethz.ch

**Jingqiu Ding**
ETH Zürich
jingqiu.ding@inf.ethz.ch

**Yiding Hua**
ETH Zürich
yiding.hua@inf.ethz.ch

**Stefan Tiegel**
ETH Zürich
stefan.tiegel@inf.ethz.ch

## Abstract

We study the problem of robustly estimating the edge density of Erdős-Rényi random graphs $\mathbb{G}(n, d^\circ/n)$ when an adversary can arbitrarily add or remove edges incident to an $\eta$-fraction of the nodes. We develop the first polynomial-time algorithm for this problem that estimates $d^\circ$ up to an additive error $O\left(\left[\sqrt{\log(n)/n} + \eta\sqrt{\log(1/\eta)}\right] \cdot \sqrt{d^\circ} + \eta \log(1/\eta)\right)$. Our error guarantee matches information-theoretic lower bounds up to factors of $\log(1/\eta)$. Moreover, our estimator works for all $d^\circ \geqslant \Omega(1)$ and achieves optimal breakdown point $\eta = 1/2$.

Previous algorithms [AJK+22, CDHS24], including inefficient ones, incur significantly suboptimal errors. Furthermore, even admitting suboptimal error guarantees, only inefficient algorithms achieve optimal breakdown point. Our algorithm is based on the sum-of-squares (SoS) hierarchy. A key ingredient is to construct constant-degree SoS certificates for concentration of the number of edges incident to small sets in $\mathbb{G}(n, d^\circ/n)$. Crucially, we show that these certificates also exist in the sparse regime, when $d^\circ = o(\log n)$, a regime in which the performance of previous algorithms was significantly suboptimal.

## 1 Introduction

We study the problem of estimating the expected average degree of Erdős-Rényi random graphs under node corruptions. The Erdős-Rényi random graph model [Gil59, ER59] is a fundamental statistical model for graphs that has been extensively studied for decades. This model has two parameters: the number of nodes ($n$) and the degree parameter ($d^\circ$) and is denoted by $\mathbb{G}(n, d^\circ/n)$. It is a distribution over graphs with $n$ nodes where each edge is sampled independently with probability $d^\circ/n$. Note that every node in $\mathbb{G}(n, d^\circ/n)$ has the same expected degree $(1 - 1/n) d^\circ$. The most fundamental statistical task in Erdős-Rényi random graphs is the following: given a graph sampled from $\mathbb{G}(n, d^\circ/n)$, find an estimate $\hat{d}$ for the ground truth parameter $d^\circ$. It is well known that the empirical average degree achieves the information-theoretically optimal error rate $|\hat{d} - d^\circ| \leqslant \Theta\left(\sqrt{\log(n)/n} \cdot \sqrt{d^\circ}\right)$.[1]

---

[1]Throughout this paper, the statistical utility guarantees we state hold with probability $1 - 1/\text{poly}(n)$ over the randomness of both the input graph and the algorithm, if not otherwise specified.

39th Conference on Neural Information Processing Systems (NeurIPS 2025).

Although many phenomena in network analysis are captured by Erdős-Rényi random graphs, the distributions of real-world networks can deviate significantly from such a basic model. This may tremendously impact the performance of algorithms that are tailored too much towards the Erdős-Rényi model. In particular, this already occurs for the very basic task of estimating the expected average degree, which is also the task we focus on in this work. Simple estimators such as the mean or the median of all degrees, or variations thereof, are known to fail drastically even when perturbing edges incident to few of the nodes [AJK⁺22]. This motivates the study of robust estimation algorithms for random graphs. Following [AJK⁺22, CDHS24] we study robust estimation for Erdős-Rényi random graphs under node corruptions, defined as follows.

**Definition 1.1** ($\eta$-corrupted Erdős-Rényi random graphs)**.** For $\eta \in [0, 1]$, an $\eta$-corrupted Erdős-Rényi random graph is generated by first sampling a graph from $\mathbb{G}(n, d^\circ/n)$, and then adversarially picking an $\eta$-fraction of the nodes and arbitrarily adding and removing edges incident to them.

Given the corruption rate $\eta$ and observation of an $\eta$-corrupted Erdős-Rényi random graph, the goal is to estimate the edge parameter $d^\circ$.

*Previous work.* While this is a seemingly simple one-dimensional robust estimation problem, in the work that initiated this line of research, [AJK⁺22] showed that many standard robust estimators, such as the median estimator, and their natural variants, such as the truncated median estimator, provably incur very suboptimal errors. On a high level, this occurs because the degrees of the uncorrupted graphs are not independent and further, because the adversary can change all of them by just corrupting a single node. On the other hand, if we do not consider any structural properties of the node corruptions and treat the problem as that of robustly estimating the parameter of a Bernoulli distribution, this can at best get error $\eta n^2$. In [AJK⁺22], they gave a polynomial-time algorithm that achieves an error rate of $O\big([\sqrt{\log(n)/n} + \eta\sqrt{\log(1/\eta)}] \cdot \sqrt{d^\circ} + \eta \log n\big)$ when $\eta < 1/60$.[3] Note that the term $\sqrt{\log n/n}\sqrt{d^\circ}$ is information-theoretically necessary even in the non-robust case. Additionally, [AJK⁺22] gave a companion lower bound, showing that information-theoretically no algorithm can achieve an error rate better than $\Omega\big(\max\{\eta\sqrt{d^\circ}, \eta\}\big)$.

*Sparse regime.* It might seem that the error rate of [AJK⁺22] is only worse than the optimal by logarithmic factors. However, because of this, it fails to provide any non-trivial statistical guarantees in the sparse regime. For example, when $d^\circ \ll \log n$ and $\eta = \Omega(1)$, their error bound is $O(\eta \log n)$ which is much larger than the ground truth parameter $d^\circ$.

Notice that, for statistical estimation problems on random graphs, it is often the case that sparse graphs are more difficult than dense graphs, such as community detection in stochastic block models [FO05, GV16, LM22, DdHS23]. When $d^\circ = o(\log n)$, Erdős-Rényi random graphs behave fundamentally different than when $d^\circ \geq \Omega(\log n)$, and usually new algorithmic and proof ideas are required. When $d^\circ \geq \Omega(\log n)$, the graph is rather regular in the sense that every vertex has degree $(1 \pm \varepsilon)d$ for a small constant $\varepsilon$, and the spectrum of the adjacency matrix concentrates nicely around its expectation: $\|A - \mathbb{E}A\| = O(\sqrt{d^\circ})$. (See e.g. [FO05].) However, when $d = o(\log n)$, significant number of vertices have degree $\omega(d)$. In particular, when $d$ is a constant, there are vertices of degree $\Omega(\log n/\log\log n)$ [KS03]. As a result, the spectral norm of the centered adjacency matrix is $\Omega(\log n/\log\log n)$.

The first step towards robust edge density estimation for sparse random graphs was made in [CDHS24]. They proposed a polynomial-time algorithm that estimates $d^\circ$ up to an additive error of $d^\circ/10$ when $\eta$ is at most some sufficiently small constant that is much smaller than $1/2$. However, their statistical guarantees are suboptimal when $d^\circ$ grows with $n$, e.g. $d^\circ = \log\log n$, since the estimator does not achieve $1 \pm o(1)$ approximation ratio for estimating $d^\circ$.

---

[2]Even in the easier corruption model in which each sample is drawn from $\mathrm{Ber}(\frac{d}{n})$ with probability $1 - \eta$ and some error distribution with probability $\eta$, it is not possible to get error better than $\eta n$ (note that the adversary can simulate this model up to constant shifts in $\eta$).

[3]Previous works [AJK⁺22, CDHS24] use a different parametrization $\mathbb{G}(n, p^\circ)$ and consider the task of estimating the edge density parameter $p^\circ$. Since $p^\circ$ and $d^\circ$ differ by a known factor of $n$, these two parameterizations are equivalent.

*Optimal breakdown point.* Besides optimal error rates, another desirable feature of robust estimation algorithms is a high breakdown point. In the node corruption model, the breakdown point of an estimator is defined to be the minimum fraction of nodes to corrupt such that the estimator cannot give any non-trivial guarantees. For the robust edge density estimation problem, it is easy to see that any estimator has breakdown point at most $1/2$, as an adversary can make $\mathbb{G}(n, 0)$ and $\mathbb{G}(n, 1)$ indistinguishable if it is allowed to corrupt half the nodes. In [AJK+22], they provide an exponential-time algorithm achieving the optimal breakdown point $1/2$ with error rate $O(\sqrt{d^\circ} + \sqrt{\log n})$.[4] Note that this error rate is quite suboptimal, as it does not recover the non-robust case when there are no corruptions, and does not provide any non-trivial guarantee in the sparse regime. On the other hand, previous polynomial-time algorithms [AJK+22, CDHS24] can provably work only when the corruption rate is at most some sufficiently small constant bounded away from $1/2$.

To summarize, current algorithms, including inefficient ones are far from the information-theoretic lower bound. This holds, even when allowing suboptimal breakdown points. Furthermore, all known efficient algorithms are both far from known lower bounds and have suboptimal breakdown points. This leads us to the following question:

> Can polynomial-time algorithms achieve error rates matching the information-theoretic lower bound? If so, can polynomial-time algorithms simultaneously achieve optimal breakdown point $1/2$?

We answer both questions affirmatively in this paper, up to factors of $\log(1/\eta)$ in the error rate.

## 1.1 Results

We give the first polynomial-time algorithm for node-robust edge density estimation in Erdős-Rényi random graphs that achieves near-optimal error rate and reaches the optimal breakdown point $1/2$.

**Theorem 1.2** (Informal restatement of Theorem C.1). *For any $0 \leqslant \eta < 1/2$ and $d^\circ \geqslant 1$, there exists a polynomial-time algorithm which, given $\eta$ and a graph that is an $\eta$-corruption of an Erdős-Rényi random graph sampled from $\mathbb{G}(n, d^\circ/n)$, outputs an estimator $\hat{d}$ satisfying*

$$|\hat{d} - d^\circ| \lesssim \left(\sqrt{\frac{\log(n)}{n}} + \eta\sqrt{\log(1/\eta)}\right) \cdot \sqrt{d^\circ} + \eta \log(1/\eta),$$

*with probability at least $1 - 1/\text{poly}(n)$.*

We make a few comments on the statistical guarantee of our algorithm. Our error rate is optimal up to the $\log(1/\eta)$ factor, as any algorithm must incur an error of $\Omega(\max\{\eta\sqrt{d^\circ}, \eta\})$ [AJK+22]. In comparison, the error rate achieved by [AJK+22] is $O\left([\sqrt{\log(n)/n} + \eta\sqrt{\log(1/\eta)}] \cdot \sqrt{d^\circ} + \eta \log n\right)$, which is worse than ours in every parameter regime. The error rate achieved by [CDHS24] is $O(d^\circ)$, which is worse when $\eta$ is small; in particular, it does not recover the non-robust case when $\eta = 0$.

The condition on $d^\circ$ can be relaxed to $d^\circ \geqslant c$ for arbitrary positive constant $c$. We remark that the condition $d^\circ \geqslant \Omega(1)$ is information-theoretically necessary if we want non-trivial guarantees for constant $\eta$, as when $d^\circ = o(1)$ most nodes will be isolated with high probability and the adversary can erase all edges, removing all information about $d^\circ$.

We leave it as an open question whether the factor of $\log(1/\eta)$ is inherently necessary, at least for polynomial-time algorithms. We remark that the lower bound in [AJK+22] is for *oblivious* adversaries, that are not allowed to see the uncorrupted graph before choosing their corruptions. Whereas we consider *adaptive* adversaries, that have full knowledge of the underlying graph. For other robust statistical inference problems, there are known separations between these models. In particular, for robustly estimating the mean of a high-dimensional

---

[4]This result does not state the dependence on $\eta$ when $\eta$ is small.

Gaussian, there are polynomial-time algorithms that achieve error $O(\eta)$ against oblivious adversaries [DKK+18].[5] But against adaptive adversaries, there are statistical query lower bound suggesting that obtaining error better than $\Omega(\eta\sqrt{\log(1/\eta)})$ takes super-polynomial time [DKS17].

As observed in [AJK+22] the key property that makes the edge density estimation problem challenging is the dependencies among the degrees of vertices. Indeed, consider the related, but much simpler, problem, in which we are given $n$ points that are an $\eta$-corruption of i.i.d. samples from a Binomial distribution with parameters $n$ and $d^\circ/n$. In this setting, the marginal distribution of each uncorrupted sample is the same as in our setting, but there are no dependencies between samples. It is not too hard to show that in this setting, the median estimator obtains error $O(\eta\sqrt{d^\circ})$ with probability at last $1 - \exp(-\eta^2 n)$ (for completeness, we give a proof of this fact in Appendix D). Our main theorem shows that we can obtain the same guarantees up to a factor of $\sqrt{\log(1/\eta)}$ in the graph setting (in most parameter regimes).

## 1.2 Notation

We introduce some notations used throughout this paper. We write $f \lesssim g$ to denote the inequality $f \leqslant C \cdot g$ for some absolute constant $C > 0$. We also use the standard asymptotic notations $O(f)$ and $\Omega(f)$ for upper and lower bounds, respectively. Random variables are denoted using boldface symbols, e.g., $\boldsymbol{X}, \boldsymbol{Y}, \boldsymbol{Z}$. For a matrix $M$, we use $\|M\|_{\mathrm{op}}$ for its spectral norm and $\|M\|_F$ for its Frobenius norm, and let $d(M)$ denote its average row/column sum, i.e. $d(M) := \sum_{i,j} M_{ij}/n$. Let $\mathbb{1}$ and $\mathbb{0}$ denote the all-one and all-zero vectors, respectively. Their dimensions will be clear from the context. We use $\|\cdot\|_2$ for the 2-norm of vectors. For any matrices (or vectors) $M, N$ of the same shape, we use $M \odot N$ to denote the element-wise product (aka Hadamard product) of $M$ and $N$. We use $G$ to denote a graph and $A = A(G)$ for its adjacency matrix, interchangeably, when the context is clear. Given a graph $G$ and a subset $S$ of nodes, let $e_G(S)$ denote the number of edges in the subgraph induced by $S$ and let $e_G(S, \bar{S})$ denote the number of edges in the cut $(S, \bar{S})$. When the graph $G$ is clear from the context, we might drop the subscript and write $e(S)$ and $e(S, \bar{S})$.

## 1.3 Organization

The rest of the paper is organized as follows. In Section 2, we give a technical overview of our results. In Appendix C, we present our main algorithm and the detailed proofs. In Appendix A, we provide some sum-of-squares background, including basic sum-of-squares proofs used in our paper. In Appendix B, we prove concentration inequalities that are used in our proofs. In Appendix D, we prove statistical guarantees of median for robust binomial mean estimation.

## 2 Techniques

Our algorithm follows the so-called *proofs-to-algorithms* framework based on the sum-of-squares (SoS) hierarchy of semidefinite programs. This framework has successfully been applied to a wide range of robust estimation tasks such as robustly estimating the mean and higher-order moments, robust linear regression, learning mixture models, and many more [KSS18, HL18, KKM18, BP21]. We refer to [RSS18] for an overview. In this framework, one first constructs a proof of identifiability of the model parameters. This already leads to an inefficient algorithm. If additionally, this proof is captured by the SoS framework, we directly obtain an efficient algorithm with the same error guarantees.

Our work thus consists of two parts: First, constructing a "simple" proof of identifiabilty and second, showing that it can be made efficient using the sum-of-squares hierarchy. We discuss the proof of identifiability in Section 2.1 and then construct the necessary SoS proof in Section 2.2 . In Section 2.3 we discuss how our approach relates to prior works. Compared

---

[5]Formally, they work for the so-called Huber contamination model.

to previous works, our proof is surprisingly simple, and we will be able to discuss it almost entirely in this section. We also view this as a strength of our work.

## 2.1 An inefficient algorithm via identifiability

The first part of our results is to find a proof of identifiability that lends itself to the "proofs-to-algorithms" paradigm. This approach is different from previous works and a key part of our work. In particular, it requires to identify a certain "goodness" condition that captures the essence of the problem. The inefficient algorithm based on this identifiability argument already surpasses the state-of-the-art among inefficient algorithms.

Denote by $G^\circ$ the uncorrupted graph and denote by $d(G^\circ)$ its empirical average degree. For brevity, denote $\delta_{\mathrm{err}} := \eta\sqrt{\log(1/\eta)}\sqrt{d^\circ} + \eta\log(1/\eta)$. Since with probability $1 - 1/\mathrm{poly}(n)$, it holds that $|d(G^\circ) - d^\circ| \leqslant O(\sqrt{\log(n)/n} \cdot \sqrt{d^\circ})$, it is enough to estimate $d(G^\circ)$ up to error $O(\delta_{\mathrm{err}})$. For the rest of this section we will focus on this task.

Our proof of identifiability follows the same line of reasoning as in many recent works on (algorithmic) robust statistics: If two datasets from some parametric distribution have first, large overlap and second, both satisfy an appropriate "goodness" condition, their underlying parameters must be close. The "goodness" condition also needs to hold for the uncorrupted dataset with high probability. This approach underlies algorithms for robust mean estimation, clustering mixture models, robust linear regression, and more.[6] This immediately yields an inefficient algorithm: Enumerate over all possible alterations of the data that still have large overlap with the input and check if they satisfy the condition. If yes, output an empirical estimator for the parameter we wish to estimate, e.g., the empirical mean.

In our setting, the datasets are graphs on $n$ nodes and large overlap refers to one graph can be obtained by arbitrarily modifying the edges incident to at most $\eta n$ nodes. We refer to two such graphs as *$\eta$-close*.

**Concentration of edges incident to small sets as goodness condition.** We next describe our goodness condition and how it leads to a proof of identifiability, and thus an inefficient algorithm. The idea behind our goodness condition is strikingly simple: Let $G$ and $G'$ be two graphs that are $\eta$-close and let $S$ be the set of node on which they disagree. We denote by $e_G(S), e_G(S, \bar{S})$ the number of edges of the subgraph induced by $S$ and the number of edges in the cut induced by $S$ (i.e., between $S$ and $\bar{S}$), respectively. When the graph $G$ is clear from context, we omit the subscript. Consider the difference in their empirical average degree,

$$\tfrac{1}{2}(d(G) - d(G')) = \tfrac{1}{n}\left(e_G(S) + e_G(S, \bar{S}) - e_{G'}(S) - e_{G'}(S, \bar{S})\right).$$

Note that the difference only depends on edges incident to the set $S$. Now, for $G$ coming from $\mathbb{G}(n, d^\circ/n)$, we know that this number is tightly concentrated around $\frac{d^\circ}{n}\binom{|S|}{2} + \frac{d^\circ}{n}|S|(n - |S|)$. In particular, this holds for *any* set $S$ of this size. If the same were true also for both $G'$, the expectation terms would cancel out and only the fluctuation remains, which would indeed be small enough. Thus, our goodness condition is exactly requiring this concentration property, replacing $d^\circ$ by the empirical average degree of the graph.[7]

Formally, denote by $N(S) = \binom{|S|}{2} + |S|(n - |S|)$, the maximum number of possible edges in the subgraph and cut induced by $S$. We require the following.

**Definition 2.1** (*$\delta$-good* graphs). Let $G$ be a graph on $n$ nodes. We say that $G$ is $\delta$-good, if for every subset $S \subseteq [n]$ of size at most $\eta n$, it holds that

$$\left| e_G(S) + e_G(S, \bar{S}) - \tfrac{d(G)}{n} \cdot N(S) \right| \leqslant \delta \cdot \eta n. \tag{2.1}$$

The parameter $\delta$ may depend on $G$.

---

[6]In many of these cases the goodness condition is bounded moments, or related conditions.
[7]This is reminiscent of the notion of *resilience*, which is the goodness condition underlying robust mean estimation for distributions with bounded moments.

Denote by $\delta(G) = O(\sqrt{\log(1/\eta)}\sqrt{d(G)} + \log(1/\eta))$.[8] By a Chernoff bound and a union bound over all sets of size at most $\eta n$, it can be shown that if $G \sim \mathbb{G}(n, d^\circ/n)$ then $G$ is $\delta(G)$-good with probability at least $1 - 1/\text{poly}(n)$ (cf. Lemma B.4). Further, $\eta\delta(G) = O(\delta_{\text{err}})$ with at least the same probability (in the latter $d(G)$ is replaced by $d^\circ$).

**Identifiability.** We next claim that this leads to identifiability: In particular, if $G$ and $G'$ are $2\eta$-close and $\delta(G)$ and $\delta(G')$ good then $|d(G) - d(G')| \leq O(\delta_{\text{err}})$. Note that this immediately gives an inefficient algorithm with error rate $O(\delta_{\text{err}})$: Simply enumerate over all graphs $\tilde{G}$ that are $\eta$-close to the input and if one of them is good, output the empirical average degree. Since this search includes the uncorrupted graph, this is successful with probability at least $1 - 1/\text{poly}(n)$. Further, any "good" graph that we find is $2\eta$-close to the uncorrupted graph. It then follows that

$$
\begin{aligned}
\tfrac{1}{2}|d(G) - d(G')| &= \tfrac{1}{n}\left|e_G(S) + e_G(S, \bar{S}) - e_{G'}(S) - e_{G'}(S, \bar{S})\right| \\
&\leq \tfrac{1}{n}\left|e_G(S) + e_G(S, \bar{S}) - \tfrac{d(G')}{n} \cdot N(S)\right| + \tfrac{1}{n}\left|e_G(S) + e_G(S, \bar{S}) - \tfrac{d(G)}{n} \cdot N(S)\right| \\
&\quad + |d(G) - d(G')| \cdot \frac{N(S)}{n^2} \,.
\end{aligned}
\tag{2.2}
$$

Since $S$ has size at most $2\eta$, it follows that $\tfrac{1}{2} - \tfrac{N(S)}{n^2} \geq \tfrac{1}{2}(1 - 2\eta)^2$. Rearranging and applying our goodness condition yields that

$$
\begin{aligned}
(1 - 2\eta)^2 |d(G) - d(G')| &\leq 2\eta\delta(G) + 2\eta\delta(G') \\
&= O\left(\eta\sqrt{\log(1/\eta)}\right) \cdot \left[\sqrt{d(G)} + \sqrt{d(G')}\right] + O\left(\eta\log(1/\eta)\right),
\end{aligned}
$$

which is the guarantee we aimed for up to the $\sqrt{d(G')}$ term. Let $\gamma = \frac{100}{(1-2\eta)^4}$. Using the AM-GM inequality and $d(G) \geq 1$[9], it follows that

$$
d(G') = \frac{d(G') - d(G)}{\sqrt{\gamma d(G)}} \cdot \sqrt{\gamma d(G)} + d(G) \leq \frac{(d(G') - d(G))^2}{\gamma} + (1 + \gamma)d(G) \,.
$$

Taking square roots and using that $\sqrt{x + y} \leq \sqrt{x} + \sqrt{y}$, it follows that

$$
\sqrt{d(G')} \leq \frac{(1 - 2\eta)^2}{10}|d(G) - d(G')| + \frac{20}{(1 - 2\eta)^2}\sqrt{d(G)} \,.
$$

Plugging this back into Eq. (2.2) yields

$$
|d(G) - d(G')| \lesssim \frac{\eta\sqrt{\log(1/\eta)} \cdot \sqrt{d(G)}}{(1 - 2\eta)^4} + \frac{\eta\log(1/\eta)}{(1 - 2\eta)^2} \,.
$$

For any $\eta$ strictly bounded away from $1/2$, this is indeed $O(\delta_{\text{err}})$, also showing that our breakdown point is $1/2$.

## 2.2 An efficient algorithm via sum-of-squares

We next describe how to design an efficient algorithm using the sum-of-squares framework. This algorithm will inherit the error rate and optimal breakdown point of the inefficient algorithm, proving Theorem 1.2.

A key part of the analysis of the inefficient algorithm was to show that the goodness condition holds with high probability for the uncorrupted graph, such that we can guarantee the exhaustive search over all $\eta$-close graphs to our input is successful. On a high level, if we additionally require that there is a *certificate* of goodness in the uncorrupted case, we can replace the exhaustive search of the inefficient algorithm by an SDP that we can solve in

---

[8]For simplicity, the reader may ignore the second part on a first read.

[9]Any other constant, potentially smaller than 1, also works.

polynomial time. Existence of the certificate then implies that this SDP is feasible. More formally, we will require that there is a constant-degree SoS proof of this fact.

It turns out that, given this certificate, we can reuse the identifiability proof above in an almost black-box way to obtain an algorithm with the same error rate. For this section, we will thus focus mainly on showing existence of this certificate. Towards the end, we will briefly explain how to use this to construct a sum-of-squares recovery algorithm. This section does not assume in-depth knowledge of the SoS proof system. We refer to Appendix A for a formal treatment.

In order to describe the certificate, we will first present an algebraic formulation of the goodness condition. Let $G \sim \mathbb{G}(n, d^\circ/n)$ be a graph with node set $[n]$ and $A$ be its adjacency matrix. For a set $S$, let $w^{(S)} \in \{0, 1\}^n$ be its indicator vector. Since it will be important later on, we will slightly switch notation and denote the empirical average degree by $d(A)$, instead of $d(G)$, to make clear that it is a function of $A$ as well. We also denote by $d_v(A)$ the degree of node $v$. Note that

$$e_G(S) + e_G(S, \bar{S}) = \sum_{v \in S} d_G(v) - e_G(S) = \sum_{v \in [n]} w_v^{(S)} d_v(A) - \tfrac{1}{2}(w^{(S)})^\top A w^{(S)}.$$

Let $p_1(w) = \sum_{v \in [n]} w_v(d_v(A) - d(A))$ and $p_2(w) = w^\top(A - \frac{d(A)}{n} \mathbb{1}\mathbb{1}^\top)w$. Then, the goodness condition holds if and only if for all $S$ of size at most $\eta n$,

$$\left(p_1(w^{(S)}) - \tfrac{1}{2} p_2(w^{(S)})\right)^2 \lesssim \delta(A)^2 \cdot (\eta n)^2, \tag{2.3}$$

where we defined $\delta(A) = O(\sqrt{\log(1/\eta)}\sqrt{d(A)} + \log(1/\eta))$ as before.[10]

The certificate is a strengthening of this fact. In particular, consider the following set of polynomial equations in variables $w_1, \ldots, w_n$: $\mathcal{A}_{\text{label}}(w; \eta) := \{\forall v : w_v^2 = w_v, \sum_v w_v \leqslant \eta n\}$ which contains all the $w^{(S)}$. We require that there is a so-called *SoS proof* of Eq. (2.3), in variables $w$. In particular, we require that the difference between right-hand side and left-hand side can be expressed as a polynomial of the form

$$\sum_r s_r(w)^2 + \sum_{i,r'} p_{i,r'}(w_i^2 - w_i) + \sum_{r''} s_{r''}(w)^2 \left(\eta n - \sum_{i=1}^n w_i\right),$$

where $s_r, p_{i,r'}, s_{r''}$ are constant-degree polynomials in $w$. We denote this by $\mathcal{A}_{\text{label}} \left|\frac{w}{O(1)}\right.$ $(p_1(w^{(S)}) - \tfrac{1}{2} p_2(w^{(S)}))^2 \lesssim \delta(A)^2 \cdot (\eta n)^2$. All of the proofs in this section will be of constant-degree, so we will drop the $O(1)$ subscript.

We will construct this in two parts. Since the SoS proof system captures the fact that $(a + b)^2 \leqslant 2a^2 + 2b^2$, it follows that $\mathcal{A}_{\text{label}} \left|\frac{w}{}\right. (p_1(w) - \tfrac{1}{2}p_2(w))^2 \leqslant 2p_1(w)^2 + p_2(w)^2$. Since SoS proofs obey composition, it is enough to certify that both $p_1^2(w)$ and $p_2(w)^2$ are at most $O(1) \cdot \delta(A)^2 \cdot (\eta n)^2$.

**Upper bounding $p_1$ via concentration of degrees on small sets.** We start by bounding $p_1$ in SoS. Note that if $w$ would correspond to a fixed point in $\mathcal{A}_{\text{label}}(w\ \eta)$, i.e., the indicator of a set of size at most $\eta n$, this would immediately follow from the standard goodness condition.

We construct the necessary SoS proof case by showing that something more general is true: Given $n$ numbers $a_1, \ldots, a_n \in \mathbb{R}$ such that for all $S \subseteq [n]$ of size at most $\eta n$ it holds that $(\sum_{v \in S} a_v)^2 \leqslant B$, there exist a constant-degree SoS proof of this fact. I.e.,

$$\mathcal{A}_{\text{label}}(w, \eta) \left|\frac{w}{O(1)}\right. \left(\sum_{v=1}^n w_v a_v\right)^2 \leqslant B.$$

On a high level, this follows by comparing the set indicated by the $w_v$ variables to the set of the $\eta n$ largest $a_v$'s. The proof only uses elementary arguments inside SoS, such as that

---

[10]Again, since it will be useful later on, we switched notation from $\delta(G)$ to $\delta(A)$.

$w_v^2 = w_v$ implies that $0 \leqslant w_v \leqslant 1$. We give the proof in Lemma A.9. The bound on $p_1$ follows as a direct corollary by picking $a_v = (\bar{d}_v(A) - \bar{d}(A))$ for $v \in V$ (where we identify $V$ with $[n]$).

**Upper bounding $p_2$ via spectral certificates.** We next turn to upper bounding $p_2$. A common strategy to show SoS upper bounds on quadratic forms is using the operator norm. Indeed, for a matrix $M$, the polynomial $\|x\|^2\|M\| - x^\top M x$ is a sum-of-squares in $x$ since the matrix $\|M\| \cdot I - M$ is positive semidefinite and can hence be written as $LL^\top$. Thus, $\|x\|^2\|M\| - x^\top M x = \|Lx\|^2$.

Unfortunately, applying this naively to bound $p_2$ does not work. Indeed, using that $\sum_{i=1}^n w_i \leqslant \eta n$ implies $\|w\|^2 \leqslant \eta n$, we can bound

$$p_2(w) = w^\top\left(A - \tfrac{d(A)}{n}\mathbb{1}\mathbb{1}^\top\right)w \leqslant \left\|A - \tfrac{d(A)}{n}\mathbb{1}\mathbb{1}^\top\right\| \cdot \eta n\,.$$

So we would need the operator norm to be on the order of $O(\sqrt{\log(1/\eta)}\sqrt{d(A)} + \log(1/\eta))$. While this is true in the dense case, when $d^\circ \geqslant \Omega(\log n)$ [FO05], this completely fails in the sparse case: In particular when $d = O(1)$, the operator norm is scales as $\sqrt{\frac{\log n}{\log\log n}} \gg \sqrt{d(A)}$ [KS03]. As a consequence, the error guarantees we would obtain would (roughly) be of the form $|\hat{d} - d^\circ| \lesssim \sqrt{\frac{\log n}{\log\log n}}$ which is very far from optimal when $d^\circ = O(1)$.

The reason for this failure is that the spectral norm is dominated by outlier nodes that have much higher degrees than $d^\circ$. Yet, there is reason for hope: The associated eigenvectors are very localized and have small correlation with the vectors of the form $w$ that we care about. Indeed, we can apply a diagonal rescaling to the centered adjacency matrix that downweighs the effect of such outlier nodes to obtain a more benign spectrum. Concretely, variants of the results in [Le16] (see also Lemma B.6 on how we need to adapt their results), show the following: Let $D \in \mathbb{R}^{n\times n}$ be the diagonal matrix, such that $D_{vv} = \max\{1, d_v(A)/(2d^\circ)\}$. Then, with probability at least $1 - 1/\mathrm{poly}(n)$, it holds that $\|D^{-1/2}(A - \tfrac{d(A)}{n}\mathbb{1}\mathbb{1}^\top)D^{-1/2}\| \lesssim \sqrt{d(A)}$.[11]

While promising, this does not come for free and we still have to ensure, that the resulting reweighing of our $w$ variables does not increase $\|w\|^2$ too much. In particular, so far we can show

$$\mathcal{A}_{\text{label}}(w;\eta) \overset{w}{\vdash} p_2(w) = w^\top\left(A - \tfrac{d(A)}{n}\mathbb{1}\mathbb{1}^\top\right)w$$

$$= \left(D^{1/2}w\right)^\top\left(D^{-1/2}\left(A - \tfrac{d(A)}{n}\mathbb{1}\mathbb{1}^\top\right)D^{-1/2}\right)\left(D^{1/2}w\right)$$

$$\leqslant \|D^{-1/2}(A - \tfrac{d(A)}{n}\mathbb{1}\mathbb{1}^\top)D^{-1/2}\| \cdot \|D^{1/2}w\|^2 \lesssim \sqrt{d(A)} \cdot \|D^{1/2}w\|^2\,. \quad (2.4)$$

It remains to bound $\|D^{1/2}w\|^2$. For this, we will use similar techniques as in the upper bound of $p_1$. Indeed, expanding the squared-norm and using that $w_v^2 = w_v$, we obtain

$$\mathcal{A}_{\text{label}}(w;\eta) \overset{w}{\vdash} \|D^{1/2}w\|^2 = \sum_{v\in V}w_v D_{vv} = \sum_{v\in V}w_v\max\left\{1, \frac{d_v(A)}{2d^\circ}\right\} \leqslant \sum_{v\in V}w_v + \frac{1}{2d^\circ}\sum_{v\in V}w_v d_v(A)$$

$$\leqslant 2\eta n + \frac{1}{2d^\circ}\sum_{v\in V}(d_v(A) - d^\circ)\,.$$

Note that the last sum is exactly the same as in $p_1$, except that $d(A)$ is replaced by $d^\circ$. This is a minor difference and we can still apply the same techniques and conclude that the above is upper bounded as

$$\eta n\left(\frac{\log(1/\eta)}{d^\circ} + \frac{\sqrt{\log(1/\eta)}}{\sqrt{d^\circ}}\right) \lesssim \eta n\left(\frac{\log(1/\eta)}{\sqrt{d^\circ}} + \sqrt{\log(1/\eta)}\right)\,.$$

---

[11]Instead of rescaling, an alternative approach to ensure better spectral behavior is to delete high-degree nodes [FO05]. We believe that this approach would likely yield an alternative certificate.

Plugging this back into Eq. (2.4) and using that $d^\circ \approx d(A)$, it follows that

$$\mathcal{A}_{\text{label}}(w;\eta) \overset{w}{\vdash} p_2(w) \lesssim \delta(G) \cdot \eta n \,.$$

Formally, we require an SoS bound on $p_2(w)^2$ instead. This can be obtained by using that similarly to the upper bound, we also have that $\overset{x}{\vdash} x^\top M x \geqslant -\|M\|\|x\|^2$. Finally, we can use that $\{-C \leqslant x \leqslant C\} \overset{x}{\vdash} x^2 \leqslant C^2$ (cf. Lemma A.8).

**Short certificates lead to efficient algorithms.** We briefly describe how to turn this into an efficient algorithm. See Appendix C for full details. In particular, consider the following constraint system in scalar-valued variables $z_v$ and matrix valued variable $Y$. $A$ is the adjacency matrix of the (corrupted) input graph. $Y$ is the "guess" of SoS for the uncorrupted graph. All but the last constraint ($\exists$ ...) ensure that $Y$ is $\eta$-close to the input graph (by $\odot$ we denote entrywise multiplication). The last constraint ensures that it satisfies the goodness condition.

$$\mathcal{A}(Y, z; A, \eta) := \begin{cases} z \odot z = z \\ \langle z, \mathbb{1} \rangle \leqslant \eta n \\ 0 \leqslant Y \leqslant \mathbb{1}\mathbb{1}^\top, \, Y = Y^\top \\ Y \odot (\mathbb{1} - z)(\mathbb{1} - z)^\top = A \odot (\mathbb{1} - z)(\mathbb{1} - z)^\top \\ \exists \, \text{SoS proof}, \\ \mathcal{A}_{\text{label}}(w; 2\eta) \overset{w}{\underset{4}{\vdash}} \langle Y - \frac{d(Y)}{n}\mathbb{1}\mathbb{1}^\top, 2w\mathbb{1}^\top - ww^\top \rangle^2 \lesssim \delta(Y)^2(\eta n)^2 \end{cases} \,. \quad (2.5)$$

The constraint $0 \leqslant Y \leqslant \mathbb{1}\mathbb{1}^\top$ is meant entry-wise. We remark that the last constraint is formally not a polynomial inequality but can be written as such using auxiliary variables. Writing this formally would very much obfuscate what is happening and we omit the details for clarity. This technique is by now standard and we refer to, e.g., [FKP+19] for more detail. On a high level, one uses auxiliary variables to search for the coefficients of the SoS proof.

The arguments in the previous section show that the uncorrupted graph corresponds to a feasible solution (cf. Lemma C.3). Further, an SoS version of the proof of identifiability we have given before shows that (cf. Lemma C.4 for the full version)

$$\mathcal{A}(Y, z; A, \eta) \overset{Y,z}{\vdash} (d(Y) - d^\circ)^2 \lesssim \eta^2 \log(1/\eta)d^\circ + \eta^2 \log^2(1/\eta) \,.$$

It then follows by known results (see, e.g., [RSS18, Theorem 2.1]) that in time $n^{O(1)}$ we can compute an estimator $\hat{d}$ such that

$$\left| \hat{d} - d^\circ \right| \lesssim \eta \sqrt{\log(1/\eta)}\sqrt{d^\circ} + \eta \log(1/\eta) \,.$$

## 2.3 Relation to previous works

While previous works do not exactly follow the general strategy to show identifiability here, they can be interpreted as implicitly trying to exploit the existence of such certificates. The crucial difference is how they try to upper bound the deviation of degrees over small subsets. In particular, they establish this by requiring that *all* degrees are tightly concentrated around $d^\circ$. This only holds when $d^\circ \gtrsim \log n$ and fails in the sparse regime. In this work, we showed that this strong concentration is not necessary. Intuitively, it is not necessary since the outlier degrees in the sparse case are washed out when averaging over the set $S$.

Besides the efficient algorithm, we also view the fact that you can apply the proofs-to-algorithms framework to this problem as a contribution of the paper. As we established, this leads to a very clean analysis and improved guarantees, that in particular also work in the sparse case and achieve optimal breakdown point.

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

# A  Sum-of-squares background

In this paper, we use the sum-of-squares (SoS) semidefinite programming hierarchy [BS14, BS16, RSS18] for both algorithm design and analysis. The sum-of-squares proofs-to-algorithms framework has been proven useful in many optimal or state-of-the-art results in algorithmic statistics [HL18, KSS18, PS17, Hop20]. We provide here a brief introduction to pseudo-distributions, sum-of-squares proofs, and sum-of-squares algorithms.

## A.1  Sum-of-squares proofs and algorithms

**Pseudo-distribution.**   We can represent a finitely supported probability distribution over $\mathbb{R}^n$ by its probability mass function $\mu \colon \mathbb{R}^n \to \mathbb{R}$ such that $\mu \geqslant 0$ and $\sum_{x \in \mathrm{supp}(\mu)} \mu(x) = 1$. We define pseudo-distributions as generalizations of such probability mass distributions by relaxing the constraint $\mu \geqslant 0$ to only require that $\mu$ passes certain low-degree non-negativity tests.

**Definition A.1** (Pseudo-distribution). A *level-$\ell$ pseudo-distribution* $\mu$ over $\mathbb{R}^n$ is a finitely supported function $\mu : \mathbb{R}^n \to \mathbb{R}$ such that $\sum_{x \in \mathrm{supp}(\mu)} \mu(x) = 1$ and $\sum_{x \in \mathrm{supp}(\mu)} \mu(x) f(x)^2 \geqslant 0$ for every polynomial $f$ of degree at most $\ell/2$.

We can define the expectation of a pseudo-distribution in the same way as the expectation of a finitely supported probability distribution.

**Definition A.2** (Pseudo-expectation). Given a pseudo-distribution $\mu$ over $\mathbb{R}^n$, we define the *pseudo-expectation* of a function $f : \mathbb{R}^n \to \mathbb{R}$ by

$$\tilde{\mathbb{E}}_{\mu} f := \sum_{x \in \mathrm{supp}(\mu)} \mu(x) f(x) \,. \tag{A.1}$$

The following definition formalizes what it means for a pseudo-distribution to satisfy a system of polynomial constraints.

**Definition A.3** (Constrained pseudo-distributions). Let $\mu : \mathbb{R}^n \to \mathbb{R}$ be a level-$\ell$ pseudo-distribution over $\mathbb{R}^n$. Let $\mathcal{A} = \{f_1 \geqslant 0, \dots, f_m \geqslant 0\}$ be a system of polynomial constraints. We say that $\mu$ *satisfies* $\mathcal{A}$ at level $r$, denoted by $\mu \models_{r} \mathcal{A}$, if for every multiset $S \subseteq [m]$ and every sum-of-squares polynomial $h$ such that $\deg(h) + \sum_{i \in S} \max\{\deg(f_i), r\} \leqslant \ell$,

$$\tilde{\mathbb{E}}_{\mu} h \cdot \prod_{i \in S} f_i \geqslant 0 \,. \tag{A.2}$$

We say $\mu$ satisfies $\mathcal{A}$ and write $\mu \models \mathcal{A}$ (without further specifying the degree) if $\mu \models_{0} \mathcal{A}$.

We remark that if $\mu$ is an actual finitely supported probability distribution, then we have $\mu \models \mathcal{A}$ if and only if $\mu$ is supported on solutions to $\mathcal{A}$.

**Sum-of-squares proof.**   We introduce sum-of-squares proofs as the dual objects of pseudo-distributions, which can be used to reason about properties of pseudo-distributions. We say a polynomial $p$ is a sum-of-squares polynomial if there exist polynomials $(q_i)$ such that $p = \sum_i q_i^2$.

**Definition A.4** (Sum-of-squares proof). A *sum-of-squares proof* that a system of polynomial constraints $\mathcal{A} = \{f_1 \geqslant 0, \dots, f_m \geqslant 0\}$ implies $q \geqslant 0$ consists of sum-of-squares polynomials $(p_S)_{S \subseteq [m]}$ such that[12]

$$q = \sum_{\text{multiset } S \subseteq [m]} p_S \cdot \prod_{i \in S} f_i \,.$$

If such a proof exists, we say that $\mathcal{A}$ *(sos-)proves* $q \geqslant 0$ within degree $\ell$, denoted by $\mathcal{A} \vdash_{\ell} q \geqslant 0$. In order to clarify the variables quantified by the proof, we often write $\mathcal{A}(x) \vdash_{\ell}^{x} q(x) \geqslant 0$.

---

[12]Here we follow the convention that $\prod_{i \in S} f_i = 1$ for $S = \emptyset$.

We say that the system $\mathcal{A}$ *sos-refuted* within degree $\ell$ if $\mathcal{A} \mathrel{\vert\kern-0.4em\raisebox{0.05em}{\tiny$\ell$}} -1 \geqslant 0$. Otherwise, we say that the system is *sos-consistent* up to degree $\ell$, which also means that there exists a level-$\ell$ pseudo-distribution satisfying the system.

The following lemma shows that sum-of-squares proofs allow us to deduce properties of pseudo-distributions that satisfy some constraints.

**Lemma A.5.** *Let $\mu$ be a pseudo-distribution, and let $\mathcal{A}, \mathcal{B}$ be systems of polynomial constraints. Suppose there exists a sum-of-squares proof $\mathcal{A} \mathrel{\vert\kern-0.4em\raisebox{0.05em}{\tiny$r'$}} \mathcal{B}$. If $\mu \mathrel{\vDash\kern-0.9em\raisebox{-0.5em}{\tiny$r$}} \mathcal{A}$, then $\mu \mathrel{\vDash\kern-1.4em\raisebox{-0.5em}{\tiny$r \cdot r' + r'$}} \mathcal{B}$.*

**Sum-of-squares algorithm.** Given a system of polynomial constraints, the *sum-of-squares algorithm* searches through the space of pseudo-distributions that satisfy this polynomial system by semidefinite programming.

Since semidefinite programing can only be solved approximately, we can only find pseudo-distributions that approximately satisfy a given polynomial system. We say that a level-$\ell$ pseudo-distribution *approximately satisfies* a polynomial system, if the inequalities in Eq. (A.2) are satisfied up to an additive error of $2^{-n^\ell} \cdot \|h\| \cdot \prod_{i \in S} \|f_i\|$, where $\|\cdot\|$ denotes the Euclidean norm[13] of the coefficients of a polynomial in the monomial basis.

**Theorem A.6** (Sum-of-squares algorithm). *There exists an $(n+m)^{O(\ell)}$-time algorithm that, given any explicitly bounded[14] and satisfiable system[15] $\mathcal{A}$ of $m$ polynomial constraints in $n$ variables, outputs a level-$\ell$ pseudo-distribution that satisfies $\mathcal{A}$ approximately.*

*Remark* A.7 (Approximation error and bit complexity). For a pseudo-distribution that only approximately satisfies a polynomial system, we can still use sum-of-squares proofs to reason about it in the same way as Lemma A.5. In order for approximation errors not to amplify throughout reasoning, we need to ensure that the bit complexity of the coefficients in the sum-of-squares proof are polynomially bounded.

## A.2 Sum-of-squares toolkit

In this part, we provide some basic SoS proofs that are useful in our paper.

**Lemma A.8.** *Given constant $C \geqslant 0$, we have*

$$\left\{ -C \leqslant x \leqslant C \right\} \mathrel{\vert\kern-0.4em\raisebox{0.05em}{\tiny$x$}_{\kern-0.6em\raisebox{-0.5em}{\tiny$2$}}} x^2 \leqslant C^2 .$$

*Proof.*

$$\left\{ -C \leqslant x \leqslant C \right\} \mathrel{\vert\kern-0.4em\raisebox{0.05em}{\tiny$x$}_{\kern-0.6em\raisebox{-0.5em}{\tiny$2$}}} (C-x)(C+x) \geqslant 0$$
$$\mathrel{\vert\kern-0.4em\raisebox{0.05em}{\tiny$x$}_{\kern-0.6em\raisebox{-0.5em}{\tiny$2$}}} C^2 - x^2 \geqslant 0$$
$$\mathrel{\vert\kern-0.4em\raisebox{0.05em}{\tiny$x$}_{\kern-0.6em\raisebox{-0.5em}{\tiny$2$}}} C^2 \geqslant x^2 .$$

$\square$

**Lemma A.9** (SoS subset sum). *Let $a_1, \ldots, a_n \in \mathbb{R}$ and $B \in \mathbb{R}$. Suppose for any subset $S \subseteq [n]$ with $|S| \leqslant k$, we have $|\sum_{i \in S} a_i| \leqslant B$. Then*

$$\left\{ 0 \leqslant x_1, \ldots, x_n \leqslant 1, \ \sum_i x_i \leqslant k \right\} \mathrel{\vert\kern-1.2em\raisebox{0.4em}{\tiny$x_1,\ldots,x_n$}_{\kern-1.8em\raisebox{-0.5em}{\tiny$1$}}} \left| \sum_{i=1}^n a_i x_i \right| \leqslant B .$$

---

[13]The choice of norm is not important here because the factor $2^{-n^\ell}$ swamps the effects of choosing another norm.

[14]A system of polynomial constraints is *explicitly bounded* if it contains a constraint of the form $\|x\|^2 \leqslant M$.

[15]Here we assume that the bit complexity of the constraints in $\mathcal{A}$ is $(n+m)^{O(1)}$.

*Proof.* We first show $\sum_{i=1}^{n} a_i x_i \leqslant B$. Without loss of generality, assume $a_1 \geqslant \ldots \geqslant a_n$.

Case 1: $a_k \geqslant 0$. It is straightforward to see $\{0 \leqslant x_1, \ldots, x_n \leqslant 1, \ \sum_i x_i \leqslant k\} \vdash_1$

$$B - \sum_{i=1}^{n} a_i x_i \geqslant \sum_{i=1}^{k} a_i - \sum_{i=1}^{n} a_i x_i \qquad (\textstyle\sum_{i=1}^{k} a_i \leqslant B)$$

$$= \sum_{i=1}^{k} a_i(1 - x_i) - \sum_{i=k+1}^{n} a_i x_i$$

$$\geqslant \sum_{i=1}^{k} a_k(1 - x_i) - \sum_{i=k+1}^{n} a_k x_i \qquad (0 \leqslant x_i \leqslant 1)$$

$$= a_k \cdot \left(k - \sum_{i=1}^{n} x_i\right)$$

$$\geqslant 0. \qquad (\textstyle\sum_i x_i \leqslant k)$$

Case 2: $a_k < 0$. Let $\ell$ be the largest index such that $a_\ell \geqslant 0$. (Note $\ell \in \{0, 1, \ldots, k-1\}$.) Then

$$B - \sum_{i=1}^{n} a_i x_i \geqslant \sum_{i=1}^{\ell} a_i - \sum_{i=1}^{n} a_i x_i \qquad (\textstyle\sum_{i=1}^{\ell} a_i \leqslant B)$$

$$= \sum_{i=1}^{\ell} a_i(1 - x_i) + \sum_{i=\ell+1}^{n} (-a_i)x_i$$

$$\geqslant 0,$$

where in the last inequality we used $a_1, \ldots, a_\ell \geqslant 0$ and $a_{\ell+1}, \ldots, a_n < 0$, as well as $0 \leqslant x_i \leqslant 1$ for all $i$.

Observe that we can apply the same argument above to $-a_1, \ldots, -a_n$ and conclude $\sum_{i=1}^{n}(-a_i)x_i \leqslant B$, or equivalently, $\sum_{i=1}^{n} a_i x_i \geqslant -B$.

Therefore,

$$\left\{0 \leqslant x_1, \ldots, x_n \leqslant 1, \ \sum_i x_i \leqslant k\right\} \vdash_1 \left\{-B \leqslant \sum_{i=1}^{n} a_i x_i \leqslant B\right\}.$$

$\square$

# B   Concentration inequalities

In this section, we prove several concentration inequalities for Erdős-Rényi random graphs that are crucially used to derive our main results. We first introduce two classical concentration bounds.

**Lemma B.1** (Chernoff bound). *Let $X_1, X_2, \ldots, X_N$ be independent random variables taking values in $\{0, 1\}$. Let $X := \sum_{i=1}^{N} X_i$. Then for any $t > 0$,*

$$\mathbb{P}(X \geqslant \mathbb{E}X + t) \leqslant \exp\left(-\frac{t^2}{t + 2\mathbb{E}X}\right),$$

$$\mathbb{P}(X \leqslant \mathbb{E}X - t) \leqslant \exp\left(-\frac{t^2}{2\mathbb{E}X}\right).$$

**Lemma B.2** (McDiarmid's inequality). *Let $X_1, X_2, \ldots, X_N$ be independent random variables. Let $f : \mathbb{R}^N \to \mathbb{R}$ be a measurable function such that the value of $f(x)$ can change by at most $c_i > 0$ under an arbitrary change of the i-th coordinate of $x \in \mathbb{R}^N$. That is, for all $x, x' \in \mathbb{R}^N$ differing only in the i-th coordinate, we have $|f(x) - f(x')| \leqslant c_i$. Then for any $t > 0$,*

$$\mathbb{P}\left(\left|f(X) - \mathbb{E}[f(X)]\right| \geqslant t\right) \leqslant 2\exp\left(-\frac{2t^2}{\sum_{i=1}^{N} c_i^2}\right).$$

**Lemma B.3** (Average degree concentration). *Let $A \sim \mathbb{G}(n, d^\circ/n)$. Let $d(A) := \frac{1}{n} \sum_{i,j} A_{ij}$. Then for every constant $C > 0$, there exists another constant $C'$ which only depends on $C$ such that*

$$\mathbb{P}\left( \left| d(A) - \mathbb{E}\, d(A) \right| \leqslant C' \cdot \max\left\{ \frac{\log n}{n}, \sqrt{\frac{\log n}{n}} \cdot \sqrt{d^\circ} \right\} \right) \geqslant 1 - n^{-C} .$$

*Proof.* Note $d(A) = \frac{2}{n} \sum_{i<j} A_{ij}$ and $\{A_{ij}\}_{i<j} \sim \mathrm{Ber}(p^\circ)$ independently. Also note $\mathbb{E}\, d(A) = (1 - 1/n)\, d^\circ$. Then by Chernoff bound Lemma B.1, for any $t > 0$,

$$\mathbb{P}\left( \left| d(A) - \mathbb{E}\, d(A) \right| \geqslant \frac{2t}{n} \right) \leqslant 2 \exp\left( -\frac{t^2}{2t + (n-1)\, d^\circ/2} \right)$$

$$\leqslant 2 \exp\left( -\frac{t^2}{2 \cdot \max\{2t, (n-1)\, d^\circ/2\}} \right)$$

$$= 2 \exp\left( -\min\left\{ \frac{t}{4}, \frac{t^2}{(n-1)\, d^\circ} \right\} \right)$$

$$= 2 \max\left\{ \exp\left( -\frac{t}{4} \right), \exp\left( -\frac{t^2}{(n-1)\, d^\circ} \right) \right\} .$$

$\square$

**Lemma B.4** (Degrees subset sum). *Let $A \sim \mathbb{G}(n, d^\circ/n)$. For $S \subseteq [n]$, let $e(S) := \sum_{i,j \in S,\, i<j} A_{ij}$ and let $e(S, \bar{S}) := \sum_{i \in S,\, j \notin S} A_{ij}$. Then for every constant $C > 0$, there exists another constant $C'$ which only depends on $C$ such that with probability $1 - n^{-C}$, we have for every $S \subseteq [n]$,*

$$\left| e(S, \bar{S}) - \mathbb{E}\, e(S, \bar{S}) \right| \leqslant C' \cdot \left( |S| \log(en/|S|) + |S| \sqrt{d^\circ \log(en/|S|)} \right) ,$$

*and*

$$\left| e(S) - \mathbb{E}\, e(S) \right| \leqslant C' \cdot \left( |S| \log(en/|S|) + |S| \sqrt{d^\circ(|S|/n) \log(en/|S|)} \right) .$$

*Proof.* Let $N \in \mathbb{N}$ and let $X_1, \ldots, X_N \sim \mathrm{Ber}(p^\circ)$ independently. Consider their sum $X := \sum_i X_i$. By Chernoff bound Lemma B.1, for any $\delta > 0$,

$$\mathbb{P}(|X - \mathbb{E}\, X| \geqslant \delta) \leqslant 2 \exp\left( -\frac{\delta^2}{\delta + 2\, \mathbb{E}\, X} \right) .$$

Fix a $k \in [n]$. By union bound, the probability that there exists a $k$-sized subset $S \subseteq [n]$ such that $\left| e(S, \bar{S}) - \mathbb{E}\, e(S, \bar{S}) \right| \geqslant \delta$ is at most

$$\binom{n}{k} \cdot 2 \exp\left( \frac{\delta^2}{\delta + 2\, \mathbb{E}\, e(S, \bar{S})} \right) \leqslant 2 \exp\left( -\frac{\delta^2}{\delta + 2\, \mathbb{E}\, e(S, \bar{S})} + k \log \frac{en}{k} \right) .$$

We want to set $\delta$ such that

$$\frac{\delta^2}{\delta + 2\, \mathbb{E}\, e(S, \bar{S})} \geqslant C \cdot k \log \frac{en}{k}$$

for some sufficiently large positive constant $C$. Then it will imply the above probability is at most

$$2 \exp\left( -(C-1) \cdot k \log \frac{en}{k} \right) \leqslant 2 \exp(-(C-1) \log(en)) \leqslant n^{-(C-1)} ,$$

where we used the fact that $x \mapsto x \log \frac{e}{x}$ is an increasing function for $x \leqslant 1$. Since

$$\frac{\delta^2}{\delta + 2\, \mathbb{E}\, e(S, \bar{S})} \geqslant \frac{1}{2} \min\left\{ \frac{\delta^2}{\delta}, \frac{\delta^2}{2\, \mathbb{E}\, e(S, \bar{S})} \right\} = \min\left\{ \frac{\delta}{2}, \frac{\delta^2}{4\, \mathbb{E}\, e(S, \bar{S})} \right\} ,$$

it suffices to ask for

$$\delta \geqslant \max\left\{ 2C \cdot k \log \frac{en}{k}, \sqrt{4C \cdot \mathbb{E}\, e(S, \bar{S}) \cdot k \log \frac{en}{k}} \right\} .$$

Then we can apply union bound to all $k = 1, \ldots, n$ and conclude that with probability at least $1 - n^{-(C-2)}$, every subset $S \subseteq [n]$ satisfies

$$\left| e(S, \bar{S}) - \mathbb{E}\, e(S, \bar{S}) \right| \lesssim |S| \log(en/|S|) + |S| \sqrt{d^\circ \log(en/|S|)}.$$

Similarly, we can replace all $e(S, \bar{S})$ by $e(S)$ in the above argument and conclude that with probability at least $1 - n^{-(C-2)}$, every subset $S \subseteq [n]$ satisfies

$$\left| e(S) - \mathbb{E}\, e(S) \right| \lesssim |S| \log(en/|S|) + |S| \sqrt{d^\circ(|S|/n) \log(en/|S|)}.$$

$\square$

**Lemma B.5.** *Let $A \sim \mathbb{G}(n, d^\circ/n)$. Then for every constant $C > 0$, there exists another constant $C'$ which only depends on $C$ such that with probability $1 - n^{-C}$, the number of vertices with degree larger than $2\,\mathbb{E}\, d(A)$ is at most $C'n/d^\circ$.*

*Proof.* Fix some $\varepsilon > 0$. (We will set $\varepsilon = 1$ in the end.) For $i \in [n]$, let $B_i$ be the $\{0,1\}$-valued indicator random variable of the event that the degree of node $i$ in $A$ is larger than $(1 + \varepsilon)\,\mathbb{E}\, d(A)$. Let $p_i := \mathbb{P}(B_i = 1)$. By Chernoff bound Lemma B.1,

$$p_i = \mathbb{P}(B_i = 1) \leq \exp\left( -\frac{\varepsilon^2\, \mathbb{E}\, d(A)}{\varepsilon + 2} \right). \tag{B.1}$$

Fix some deviation $\delta > 0$. We are going to upper bound $\mathbb{P}(\sum_i B_i - \mathbb{E} \sum_i B_i \geq \delta)$. To this end, consider the $n$-stage vertex exposure martingale (note $B_i$'s are not independent) where at the $i$th stage we reveal all edges incident to the first $i$ nodes. Formally, let $S_i := \{A_{ij}\}_{j>i}$ (note $S_i$'s are independent) and define $f(S_1, \ldots, S_n) := \sum_i B_i$. However, the Lipschitz constant of $f$ is $\Omega(n)$ which is too large for us to apply McDiarmid's inequality. To reduce the Lipschitz constant, we introduce a truncation function $t$ as follows. Fix a bound $\Delta \geq 1$. For $i \in [n]$, let $t(S_i) = S_i$ if there are at most $\Delta$ ones in $S_i$; otherwise, set $t(S_i) = 0$. Let $g(S_1, \ldots, S_n) := f(t(S_1), \ldots, t(S_n))$. Then it is straightforward to see the Lipschitz constant of $g$ is at most $3\Delta$. Then by McDiarmid's inequality Lemma B.2,

$$\mathbb{P}(g - \mathbb{E}\, g \geq \delta) \leq \exp\left( -\frac{2\delta^2}{9n\Delta^2} \right).$$

To make this probability at most $n^{-C}$, it suffices to set $\delta \geq \sqrt{9C/2}\, \Delta \sqrt{n \log n}$.

Then we relate $f$ and $g$. By the definition of $g$, we have $f(S_1, \ldots, S_n) \geq g(S_1, \ldots, S_n)$ with probability 1, which implies $\mathbb{E}\, f \geq \mathbb{E}\, g$. By Chernoff bound and union bound, there exists a constant $C' > 0$ which only depends on $C$ such that the maximum degree of $A$ is at most $d^\circ + C' \cdot (\sqrt{d^\circ \log n} + \log n)$ with probability $1 - n^{-C}$. Thus, we set $\Delta = d^\circ + C' \cdot (\sqrt{d^\circ \log n} + \log n)$. Then we have $f(S_1, \ldots, S_n) = g(S_1, \ldots, S_n)$ with probability $1 - n^{-C}$.

Putting things together, by union bound, $f = g$ and $g - \mathbb{E}\, g \leq \delta$ happen simultaneously with probability $1 - 2n^{-C}$. Together with $\mathbb{E}\, f \geq \mathbb{E}\, g$, they imply

$$f - \mathbb{E}\, f \leq \sqrt{9C/2} \cdot \sqrt{n \log n}\left( d^\circ + C'\sqrt{d^\circ \log n} + C' \log n \right)$$

with probability $1 - 2n^{-C}$.

Finally, plugging in $\varepsilon = 1$ to Eq. (B.1), we have the expectation of the number of nodes with degree larger than $2\,\mathbb{E}\, d(A)$ is upper bounded by

$$ne^{-\mathbb{E}\, d(A)/3} \leq 2n/\mathbb{E}\, d(A),$$

where we used $e^{-x/3} \leq 2/x$ for all $x > 0$. To ensure the deviation at most $O(n/d^\circ)$, it suffices to require

$$\sqrt{n \log n}\left( d^\circ + \sqrt{d^\circ \log n} + \log n \right) \lesssim \frac{n}{d^\circ} \iff d^\circ \lesssim (n/\log n)^{1/4}.$$

Note when $d^\circ \gtrsim (n/\log n)^{1/4}$, it is easy to see the maximum degree of $A$ is $(1 + o(1))d^\circ$ with probability $1 - 1/\mathrm{poly}(n)$. $\square$

**Lemma B.6.** *Let $A \sim \mathbb{G}(n, d^\circ/n)$. Let $\tilde{D}$ be the n-by-n diagonal matrix whose i-th diagonal entry is $\max\{1, d_i/2\mathbb{E}\,d(A)\}$ where $d_i$ is the degree of node i. Then with probability $1 - 1/\text{poly}(n)$,*

$$\left\|\tilde{D}^{-1/2}(A - \mathbb{E}\,A)\tilde{D}^{-1/2}\right\|_{\text{op}} \lesssim \sqrt{d^\circ}\,.$$

*Proof.* First, let $T \subseteq [n]$ be the index set corresponding to vertices of degree larger than $2\mathbb{E}\,d(A)$. By Lemma B.5, we know that $|T| \lesssim n/d^\circ$ with probability at least $1 - 1/\text{poly}(n)$. We condition on this event. Note that this implies that the rescaling by $\tilde{D}^{-1/2}$ affects at most $O(n/d^\circ)$ vertices. By [Le16, Theorem 2.2.1] (see also Point 4 in Section 2.1.4), it follows that $\left\|\tilde{D}^{-1/2}A\tilde{D}^{-1/2} - \mathbb{E}\,A\right\|_{\text{op}} \lesssim \sqrt{d^\circ}$ with probability at least $1 - 1/\text{poly}(n)$. It is thus enough to show that with the same probability $\left\|\mathbb{E}\,A - \tilde{D}^{-1/2}\,\mathbb{E}[A]\,\tilde{D}^{-1/2}\right\|_{\text{op}} \lesssim \sqrt{d^\circ}$. Note that $\mathbb{E}\,A = \frac{d^\circ}{n}(\mathbb{1}\mathbb{1}^\top - I_n)$. Throughout the proof we will pretend it is equal to $\frac{d^\circ}{n}\mathbb{1}\mathbb{1}^\top$, it can be easily checked that the difference is a lower order term. Let $c = \tilde{D}^{-1/2}\mathbb{1}$. We will show that $\|cc^\top - \mathbb{1}\mathbb{1}^\top\|_{\text{op}} \lesssim \frac{n}{\sqrt{d^\circ}}$ with probability at least $1 - 1/\text{poly}(n)$, which implies the claim.

Note that for $i \notin T$ it holds that $c_i = 1$. Further, for all $i$, we have $0 \leqslant c_i \leqslant 1$. Thus,

$$\left\|cc^\top - \mathbb{1}\mathbb{1}^\top\right\|_{\text{op}}^2 \leqslant \left\|cc^\top - \mathbb{1}\mathbb{1}^\top\right\|_{\text{F}}^2 \leqslant 2 \sum_{i \in T, j \in [n]} \left(1 + c_i^2 c_j^2\right) \leqslant 4n|T| \lesssim \frac{n^2}{d^\circ}\,.$$

$\square$

# C  Robust edge density estimation algorithm

In this section, we show that there exists an algorithm that achieves the following guarantee.

**Theorem C.1.** *Given corruption rate $\eta \in [0, \frac{1}{2})$ and the $\eta$-corrupted adjacency matrix $A$ of an Erdős-Rényi random graph $A^\circ \sim \mathbb{G}(n, d^\circ/n)$ where $d^\circ \geqslant c$ for any constant $c > 0$, there exists a polynomial-time algorithm that outputs estimator $\hat{d}$ that satisfies*

$$\left|\hat{d} - d^\circ\right| \lesssim \sqrt{\frac{\log(n) \cdot d^\circ}{n}} + \frac{\eta\sqrt{\log(1/\eta) \cdot d^\circ}}{(1 - 2\eta)^4} + \frac{\eta\log(1/\eta)}{(1 - 2\eta)^2}\,,$$

*with probability $1 - 1/\text{poly}(n)$.*

*Remark C.2.* When $\eta \leqslant \frac{1}{2} - \varepsilon$ for any constant $\varepsilon \in (0, 1/2]$, the error bound becomes

$$\left|\hat{d} - d^\circ\right| \lesssim \sqrt{\frac{\log(n) \cdot d^\circ}{n}} + \eta\sqrt{\log(1/\eta) \cdot d^\circ} + \eta\log(1/\eta)\,.$$

The main idea of the algorithm follows the general paradigm of robust statistics via sum-of-squares: given observation of the $\eta$-corrupted graph $A$, find a graph $Y$ that differs from $A$ by at most $\eta n$ vertices and satisfies a set of properties $\mathcal{P}$ of the uncorrupted Erdős-Rényi graph $A^\circ \sim \mathbb{G}(n, d^\circ/n)$, then output the average degree $d(Y)$ as the estimator.

The crux of the algorithm is to determine the properties $\mathcal{P}$ that $Y$ satisfies such that:

- $\mathcal{P}$ is sufficient to show that $d(Y)$ is provably close to the $d^\circ$.
- $\mathcal{P}$ is efficiently certifiable in SoS.

In our algorithm, this paradigm is implemented by the following three polynomial systems with variables $Y = (Y_{ij})_{i,j \in [n]}$ and $z = (z_i)_{i \in [n]}$ and with inputs the corruption rate $\eta$ and $\eta$-corrupted adjacency matrix $A$.

**Integrality and size constraint of labeling.** The first polynomial system guarantees that $z$ is integral and sums up to at most $\eta n$.

$$\mathcal{A}_{\text{label}}(z; \eta) := \left\{z \odot z = z, \langle \mathbb{1}, z \rangle \leqslant \eta n\right\}. \tag{C.1}$$

**Graph agreement constraint.** The second polynomial system is to certify that $Y$ differs from the observed $\eta$-corrupted graph $A$ by at most $\eta n$ vertices.

$$\mathcal{A}_{\text{graph}}(Y, z; A, \eta) := \begin{cases} 0 \leqslant Y \leqslant \mathbb{1}\mathbb{1}^{\top}, Y = Y^{\top} \\ Y \odot (\mathbb{1} - z)(\mathbb{1} - z)^{\top} = A \odot (\mathbb{1} - z)(\mathbb{1} - z)^{\top} \end{cases}. \tag{C.2}$$

**Sum-of-squares certification of degree concentration.** The third polynomial system is the property $\mathcal{P}$ that is efficiently certifiable and guarantees that $d(Y)$ will be provably close to $d^{\circ}$.

$$\mathcal{A}_{\text{degree}}(Y; \eta) := \begin{cases} \exists \text{ SoS proof in } w \text{ that,} \\ \mathcal{A}_{\text{label}}(w; 2\eta) \left|\frac{w}{4}\right. \langle Y - \frac{d(Y)}{n}\mathbb{1}\mathbb{1}^{\top}, 2w\mathbb{1}^{\top} - ww^{\top} \rangle^2 \leqslant C_1(\eta) \cdot n^2 \cdot d(Y) + C_2(\eta) \cdot n^2 \end{cases}, \tag{C.3}$$

where $C_1(\eta) = C_{\text{deg}}\eta^2 \log(1/\eta)$, $C_2(\eta) = C_{\text{deg}}\eta^2 \log^2(1/\eta)$, and $C_{\text{deg}}$ is a universal constant. Note that the SoS certificate constraint $\mathcal{A}_{\text{degree}}(Y; \eta)$ can be formally modeled using polynomial inequalities by introducing auxiliary variables for the SoS proof coefficients. This is a standard technique and we refer the reader to [KSS18, FKP+19] for more detailed discussions.

For the convenience of notation, we will consider the following combined polynomial system in remaining of the section

$$\mathcal{A}(Y, z; A, \eta) := \mathcal{A}_{\text{label}}(z; \eta) \cup \mathcal{A}_{\text{graph}}(Y, z; A, \eta) \cup \mathcal{A}_{\text{degree}}(Y, R; \eta). \tag{C.4}$$

We will show that $\mathcal{A}(Y, z; A, \eta)$ is both feasible and provides meaningful guarantees.

**Feasibility.** It can be shown that $\mathcal{A}(Y, z; A, \eta)$ is satisfied by uncorrupted Erdős-Rényi random graphs with high probability. More concretely, we will prove the following feasibility lemma in Appendix C.1.

**Lemma C.3** (Feasibility). *Let $A^{\circ} \sim \mathbb{G}(n, d^{\circ}/n)$ with $d^{\circ} \geqslant c$ for any constant $c > 0$. Let $\eta \in [0, 1/2)$ and $A$ be an $\eta$-corrupted version of $A^{\circ}$. With probability $1 - 1/\text{poly}(n)$, there exists $z^{\circ} \in \{0, 1\}^n$ such that $(Y, z) = (A^{\circ}, z^{\circ})$ is a feasible solution to $\mathcal{A}(Y, z; A, \eta)$.*

**Utility.** It can be shown that any pairs of feasible solutions $(Y^*, z^*)$ and $(Y, z)$ to low-degree SoS relaxation of $\mathcal{A}(Y, z; A, \eta)$ satisfy that $|d(Y) - d(Y^*)|$ is sufficiently small. More concretely, we will prove the following utility lemma in Appendix C.2.

**Lemma C.4** (Utility). *Given $A \in \{0, 1\}^{n \times n}$ and constant $\eta$ such that $\eta \in [0, 1/2)$, if $(Y^*, z^*)$ is a feasible solution to $\mathcal{A}(Y, z; A, \eta)$ and $d(Y^*) \geqslant C_{Y^*}$ for any constant $C_{Y^*} > 0$, then*

$$\mathcal{A} \left|\frac{Y, z}{8}\right. (d(Y) - d(Y^*))^2 \lesssim \frac{\eta^2 \log(1/\eta)}{(1 - 2\eta)^8} d(Y^*) + \frac{\eta^2 \log^2(1/\eta)}{(1 - 2\eta)^4}.$$

Now, we are ready to describe our algorithm that satisfy Theorem C.1. The algorithm computes the degree-8 pseudo-expectation $\tilde{\mathbb{E}}$ of $\mathcal{A}(Y, z; A, \eta)$ by solving the level-8 SoS relaxation of the integer program in Eq. (C.4), then it outputs $\tilde{\mathbb{E}}[d(Y)]$ as the estimator $\hat{d}$.

---

**Algorithm C.5** (Robust edge density estimation).
**Input:** $\eta$-corrupted adjacency matrix $A$ and corruption fraction $\eta$.

**Algorithm:** Obtain degree-8 pseudo-expectation $\tilde{\mathbb{E}}$ by solving level-8 sum-of-squares relaxation of program $\mathcal{A}(Y, z; A, \eta)$ (defined in Eq. (C.4)) with input $A$ and $\eta$.

**Output:** $\tilde{\mathbb{E}}[d(Y)]$.

---

*Proof of Theorem C.1.* We will show that Algorithm C.5 satisfies the guarantees of Theorem C.1.

By Lemma C.3, we know that, with probability $1 - 1/\text{poly}(n)$, $\mathcal{A}(Y, z; A, \eta)$ is satisfied by $(Y, z) = (A^{\circ}, z^{\circ})$ where $A^{\circ} \sim \mathbb{G}(n, d^{\circ}/n)$ is the uncorrupted graph and $z^{\circ}$ is the set of corrupted vertices.

By Lemma B.3, the average degree of $A^\circ \sim \mathbb{G}(n, d^\circ/n)$ satisfies, with probability $1 - 1/\mathrm{poly}(n)$,

$$|d(A^\circ) - d^\circ| \lesssim \sqrt{\frac{d^\circ \log(n)}{n}} . \tag{C.5}$$

Therefore, $d(A^\circ) \geq d^\circ - \sqrt{\frac{d^\circ \log(n)}{n}} \geq c - o(1) \geq C_{A^\circ}$.

Now, we can apply Lemma C.4 with $(Y^*, z^*) = (A^\circ, z^\circ)$ and $C_{Y^*} = C_{A^\circ}$. This implies that the degree-8 pseudo-expectation $\tilde{\mathbb{E}}$ obtained in Algorithm C.5 satisfies

$$\tilde{\mathbb{E}}[(d(Y) - d(A^\circ))^2] \lesssim \frac{\eta^2 \log(1/\eta)}{(1 - 2\eta)^8} d(A^\circ) + \frac{\eta^2 \log^2(1/\eta)}{(1 - 2\eta)^4} .$$

By Jensen's inequality for pseudo-expectations, it follows that

$$\left(\tilde{\mathbb{E}}[d(Y)] - d(A^\circ)\right)^2 \lesssim \frac{\eta^2 \log(1/\eta)}{(1 - 2\eta)^8} d(A^\circ) + \frac{\eta^2 \log^2(1/\eta)}{(1 - 2\eta)^4} .$$

Since the estimator output by Algorithm C.5 is $\hat{d} = \tilde{\mathbb{E}}[d(Y)]$, it follows that

$$\left|\hat{d} - d(A^\circ)\right| \lesssim \sqrt{\frac{\eta^2 \log(1/\eta)}{(1 - 2\eta)^8} d(A^\circ) + \frac{\eta^2 \log^2(1/\eta)}{(1 - 2\eta)^4}}$$

$$\leq \frac{\eta \sqrt{\log(1/\eta)}}{(1 - 2\eta)^4} \sqrt{d(A^\circ)} + \frac{\eta \log(1/\eta)}{(1 - 2\eta)^2} .$$

By Eq. (C.5), we have $d(A^\circ) \leq \sqrt{\frac{d^\circ \log(n)}{n}} + d^\circ \leq 2d^\circ$ with probability $1 - 1/\mathrm{poly}(n)$. Therefore,

$$\left|\hat{d} - d(A^\circ)\right| \lesssim \frac{\eta \sqrt{\log(1/\eta)}}{(1 - 2\eta)^4} \sqrt{d^\circ} + \frac{\eta \log(1/\eta)}{(1 - 2\eta)^2} .$$

By triangle inequality and Eq. (C.5), we have

$$\left|\hat{d} - d^\circ\right| \leq \left|\hat{d} - d(A^\circ)\right| + \left|d(A^\circ) - d^\circ\right| \lesssim \sqrt{\frac{\log(n) \cdot d^\circ}{n}} + \frac{\eta \sqrt{\log(1/\eta) \cdot d^\circ}}{(1 - 2\eta)^4} + \frac{\eta \log(1/\eta)}{(1 - 2\eta)^2} .$$

By union bound on failure probability of Lemma C.3 and Lemma B.3, Algorithm C.5 succeeds with probability $1 - 1/\mathrm{poly}(n)$. This finishes the proof for the error guarantee.

For time complexity of Algorithm C.5, since Eq. (C.4) has polynomial bit complexity, solving the level-8 SoS relaxation of Eq. (C.4) and evaluating $\tilde{\mathbb{E}}[d(Y)]$ can be done in polynomial time. Therefore, Theorem C.1 has polynomial runtime, which finishes the proof. □

## C.1 SoS feasibility

In this section, we prove Lemma C.3, which is a direct corollary of the following lemma.

**Lemma C.6.** *For any constants $c, C > 0$, the following holds. Let $A \sim \mathbb{G}(n, d^\circ/n)$ with $d^\circ \geq c$. Let $d(A) = \frac{1}{n} \sum_{i,j} A_{ij}$ and $\gamma \in [0, 1]$. Then with probability at least $1 - n^{-C}$, we have $\{w \odot w = w, \langle \mathbb{1}, w \rangle \leq \gamma n\} \left|\frac{w}{4}\right.$*

$$\left\langle A - \frac{d(A)}{n} \mathbb{1}\mathbb{1}^\top, 2w\mathbb{1}^\top - ww^\top \right\rangle^2 \leq C' \cdot \left( \gamma^2 \log(e/\gamma) n^2 \cdot d(A) + \gamma^2 \log^2(e/\gamma) n^2 \right),$$

*where $C'$ only depends on $c$ and $C$.*

*Proof.* Note

$$\left\langle A - \frac{d(A)}{n}\mathbb{1}\mathbb{1}^\top,\ 2w\mathbb{1}^\top - ww^\top \right\rangle = 2 \cdot \left\langle w, A\mathbb{1} - d(A)\cdot\mathbb{1} \right\rangle - w^\top\left( A - \frac{d(A)}{n}\mathbb{1}\mathbb{1}^\top \right)w\,.$$

We will bound $\langle w, A\mathbb{1} - d(A)\cdot\mathbb{1}\rangle$ and $w^\top(A - \frac{d(A)}{n}\mathbb{1}\mathbb{1}^\top)w$ separately. By Lemma B.3, we have

$$\left| d(A) - \mathbb{E}\,d(A) \right| \lesssim \max\left\{ \frac{\log n}{n},\ \sqrt{\frac{\log n}{n}}\cdot\sqrt{d^\circ} \right\}$$

with probability $1 - 1/\mathrm{poly}(n)$. (Since $d^\circ \geqslant \Omega(1)$, this bound can be simplified to $|d(A) - \mathbb{E}\,d(A)| \lesssim \sqrt{\log(n)/n}\cdot\sqrt{d^\circ}$.) We will condition our following analysis on this event.

**Bounding $\langle w, A\mathbb{1} - d(A)\cdot\mathbb{1}\rangle$.** Note

$$\langle w, A\mathbb{1} - d(A)\cdot\mathbb{1}\rangle = \langle w, A\mathbb{1} - \mathbb{E}\,d(A)\cdot\mathbb{1}\rangle + \left(\mathbb{E}\,d(A) - d(A)\right)\cdot\langle\mathbb{1},w\rangle\,, \tag{C.6}$$

where the second term on the right side can be easily bounded as follows, $\{\mathbb{0} \leqslant \langle\mathbb{1},w\rangle \leqslant \gamma n\}\ \Big|_{\overline{1}}$

$$\left|\left(\mathbb{E}\,d(A) - d(A)\right)\cdot\langle\mathbb{1},w\rangle\right| \leqslant \left|\mathbb{E}\,d(A) - d(A)\right|\cdot\gamma n \lesssim \gamma\cdot\max\left\{\log n,\ \sqrt{n\log(n)\,d^\circ}\right\}\,. \tag{C.7}$$

Now we bound the first term on the right side of Eq. (C.6). For $i \in [n]$, let $d_i$ be the degree of node $i$ in $A$. Then

$$\langle w, A\mathbb{1} - \mathbb{E}\,d(A)\cdot\mathbb{1}\rangle = \sum_i w_i(d_i - \mathbb{E}\,d(A))\,.$$

For a subset $S \subseteq [n]$, we have $\sum_{i\in S} d_i = e(S,\bar{S}) + 2e(S)$, which implies

$$\sum_{i\in S}(d_i - \mathbb{E}\,d(A)) = e(S,\bar{S}) + 2e(S) - \mathbb{E}\,e(S,\bar{S}) - 2\,\mathbb{E}\,e(S)\,.$$

Hence, by Lemma B.4, the following holds with probability $1 - 1/\mathrm{poly}(n)$: For every subset $S \subseteq [n]$ with $|S| \leqslant \gamma n$,

$$\left|\sum_{i\in S}(d_i - \mathbb{E}\,d(A))\right| \leqslant \left|e(S,\bar{S}) - \mathbb{E}\,e(S,\bar{S})\right| + 2\left|e(S) - \mathbb{E}\,e(S)\right|$$

$$\lesssim \gamma\log(e/\gamma)\,n + \gamma\sqrt{\log(e/\gamma)}\,n\sqrt{d^\circ}\,.$$

By Lemma A.9, the above is also true in SoS, i.e., $\{\mathbb{0} \leqslant w \leqslant \mathbb{1},\ \langle\mathbb{1},w\rangle \leqslant \gamma n\}\ \Big|_{\overline{1}}$

$$\left|\sum_i w_i(d_i - \mathbb{E}\,d(A))\right| \lesssim \gamma\log(e/\gamma)\,n + \gamma\sqrt{\log(e/\gamma)}\,n\sqrt{d^\circ}\,. \tag{C.8}$$

Therefore, putting Eq. (C.7) and Eq. (C.8) together, we have $\{\mathbb{0} \leqslant w \leqslant \mathbb{1},\ \langle\mathbb{1},w\rangle \leqslant \gamma n\}\ \Big|_{\overline{1}}$

$$\left|\langle w, A\mathbb{1} - d(A)\cdot\mathbb{1}\rangle\right| \lesssim \gamma\log(e/\gamma)\,n + \gamma\sqrt{\log(e/\gamma)}\,n\sqrt{d^\circ} \tag{C.9}$$

with probability $1 - 1/\mathrm{poly}(n)$.

**Bounding $w^\top(A - \frac{d(A)}{n}\mathbb{1}\mathbb{1}^\top)w$.** Note

$$w^\top\left(A - \frac{d(A)}{n}\mathbb{1}\mathbb{1}^\top\right)w = w^\top\left(A - \mathbb{E}\,A\right)w + w^\top\left(\mathbb{E}\,A - \frac{d(A)}{n}\mathbb{1}\mathbb{1}^\top\right)w\,. \tag{C.10}$$

We bound the first term on the right hand side of Eq. (C.10). Let $\tilde{D}$ be the $n$-by-$n$ diagonal matrix whose $i$-th diagonal entry is $\max\{1, d_i/2\,\mathbb{E}\,d(A)\}$. As

$$w^\top(A - \mathbb{E}\,A)w = (\tilde{D}^{1/2}w)^\top\,\tilde{D}^{-1/2}(A - \mathbb{E}\,A)\tilde{D}^{-1/2}\,(\tilde{D}^{1/2}w)\,,$$

then
$$\left|w^\top(A - \mathbb{E}\,A)w\right| \leqslant \left\|\tilde{D}^{-1/2}(A - \mathbb{E}\,A)\tilde{D}^{-1/2}\right\|_{\mathrm{op}}\left\|\tilde{D}^{1/2}w\right\|_2^2.$$

By Lemma B.6, the following holds with probability $1 - 1/\mathrm{poly}(n)$:
$$\left\|\tilde{D}^{-1/2}(A - \mathbb{E}\,A)\tilde{D}^{-1/2}\right\|_{\mathrm{op}} \lesssim \sqrt{d^\circ}.$$

Since
$$\{0 \leqslant w \leqslant 1\} \;\Big|\!\frac{}{2}\; \left\|\tilde{D}^{1/2}w\right\|_2^2 = \sum_i \max\left\{1, \frac{d_i}{2\,\mathbb{E}\,d(A)}\right\}w_i^2$$
$$\leqslant \sum_i w_i + \sum_i \frac{d_i}{2\,\mathbb{E}\,d(A)}w_i$$
$$= \frac{3}{2}\sum_i w_i + \frac{1}{2\,\mathbb{E}\,d(A)}\sum_i w_i(d_i - \mathbb{E}\,d(A)),$$

then by Eq. (C.8),
$$\{0 \leqslant w \leqslant 1,\ \langle 1, w\rangle \leqslant \gamma n\} \;\Big|\!\frac{}{2}\; \left\|\tilde{D}^{1/2}w\right\|_2^2 \lesssim \gamma n + \frac{\gamma \log(e/\gamma)\,n}{d^\circ} + \frac{\gamma\sqrt{\log(e/\gamma)}\,n}{\sqrt{d^\circ}}.$$

Therefore, the following holds with probability $1 - 1/\mathrm{poly}(n)$: $\{0 \leqslant w \leqslant 1,\ \langle 1, w\rangle \leqslant \gamma n\} \;\Big|\!\frac{}{2}$
$$\left|w^\top(A - \mathbb{E}\,A)w\right| \lesssim \gamma n\sqrt{d^\circ} + \frac{\gamma\log(e/\gamma)\,n}{\sqrt{d^\circ}} + \gamma\sqrt{\log(e/\gamma)}\,n. \tag{C.11}$$

Now we bound the second term on the right hand side of Eq. (C.10). Since
$$\mathbb{E}\,A - \frac{d(A)}{n}11^\top = \frac{d^\circ}{n}\left(11^\top - I\right) - \frac{d(A)}{n}11^\top$$
$$= \frac{d^\circ - d(A)}{n}11^\top - \frac{d^\circ}{n}I,$$

then
$$w^\top\left(\mathbb{E}\,A - \frac{d(A)}{n}11^\top\right)w = \frac{d^\circ - d(A)}{n}\langle 1, w\rangle^2 - \frac{d^\circ}{n}\|w\|_2^2.$$

Thus, $\{0 \leqslant w \leqslant 1,\ \langle 1, w\rangle \leqslant \gamma n\} \;\Big|\!\frac{}{2}$
$$\left|w^\top\left(\mathbb{E}\,A - \frac{d(A)}{n}11^\top\right)w\right| \leqslant \gamma^2 n\left|d^\circ - d(A)\right| + \gamma d^\circ$$
$$\lesssim \gamma^2 \log n + \gamma^2\sqrt{n\log(n)\,d^\circ} + \gamma d^\circ. \tag{C.12}$$

Therefore, putting Eq. (C.11) and Eq. (C.12) together, and assuming $d^\circ \geqslant \Omega(1)$, we have
$$\{0 \leqslant w \leqslant 1,\ \langle 1, w\rangle \leqslant \gamma n\} \;\Big|\!\frac{}{2}\; \left|w^\top\left(A - \frac{d(A)}{n}11^\top\right)w\right| \lesssim \gamma\log(e/\gamma)\,n + \gamma\sqrt{\log(e/\gamma)}\,n\sqrt{d^\circ}$$
$$\tag{C.13}$$

with probability $1 - 1/\mathrm{poly}(n)$.

**Putting things together.** Using two simple SoS facts $\Big|\!\frac{x,y}{}\; \{(x+y)^2 \leqslant 2x^2 + 2y^2\}$ and $\{|x| \leqslant B\} \;\Big|\!\frac{x}{}\; \{x^2 \leqslant B^2\}$, together with Eq. (C.9) and Eq. (C.13), we have
$$\left\langle A - \frac{d(A)}{n}11^\top,\ 2w1^\top - ww^\top\right\rangle^2 = \left(2\langle w, A1 - d(A)\cdot 1\rangle - w^\top\left(A - \frac{d(A)}{n}11^\top\right)w\right)^2$$
$$\leqslant 8\langle w, A1 - d(A)\cdot 1\rangle^2 + 2\left(w^\top\left(A - \frac{d(A)}{n}11^\top\right)w\right)^2$$
$$\lesssim \gamma^2\log(e/\gamma)n^2 d^\circ + \gamma^2\log^2(e/\gamma)n^2$$
$$\lesssim \gamma^2\log(e/\gamma)n^2 d(A) + \gamma^2\log^2(e/\gamma)n^2,$$

where in the last step we used $d(A) = (1 + o(1))d^\circ$. $\qquad\square$

## C.2  SoS utility

In this section, we prove Lemma C.4. The key observation is that $Y$ and $Y^*$ will have large agreement because they both agree with $A$ on $(1-\eta)n$ vertices. Therefore, $|d(Y) - d(Y^*)|$ only depends the set of vertices of size at most $2\eta n$ that $Y$ and $Y^*$ differ, which can be bounded by the SoS certificate in $\mathcal{A}_{\text{degree}}(Y;\eta)$.

*Proof of Lemma C.4.* Let $w = \mathbb{1} - (\mathbb{1}-z)\odot(\mathbb{1}-z^*)$. By constraints $Y\odot(\mathbb{1}-z)(\mathbb{1}-z)^\top = A\odot(\mathbb{1}-z)(\mathbb{1}-z)^\top$ and $Y^*\odot(\mathbb{1}-z^*)(\mathbb{1}-z^*)^\top = A\odot(\mathbb{1}-z^*)(\mathbb{1}-z^*)^\top$, we have

$$
\mathcal{A}\;\Big|\frac{Y,z}{4}\; Y\odot(\mathbb{1}-w)(\mathbb{1}-w)^\top = Y\odot(\mathbb{1}-z)(\mathbb{1}-z)^\top\odot(\mathbb{1}-z^*)(\mathbb{1}-z^*)^\top
$$
$$
= A\odot(\mathbb{1}-z)(\mathbb{1}-z)^\top\odot(\mathbb{1}-z^*)(\mathbb{1}-z^*)^\top
$$
$$
= A\odot(\mathbb{1}-z^*)(\mathbb{1}-z^*)^\top\odot(\mathbb{1}-z)(\mathbb{1}-z)^\top
$$
$$
= Y^*\odot(\mathbb{1}-z^*)(\mathbb{1}-z^*)^\top\odot(\mathbb{1}-z)(\mathbb{1}-z)^\top
$$
$$
= Y^*\odot(\mathbb{1}-w)(\mathbb{1}-w)^\top .
$$

Therefore, it follows that

$$
\mathcal{A}\;\Big|\frac{Y,z}{8}\; n\Big(d(Y)-d(Y^*)\Big) = \langle Y-Y^*, \mathbb{1}\mathbb{1}^\top\rangle
$$
$$
= \langle Y-Y^*, \mathbb{1}\mathbb{1}^\top - (\mathbb{1}-w)(\mathbb{1}-w)^\top\rangle
$$
$$
= \langle Y-Y^*, 2w\mathbb{1}^\top - ww^\top\rangle
$$
$$
= \langle Y-\frac{d(Y)}{n}\mathbb{1}\mathbb{1}^\top + \frac{d(Y)}{n}\mathbb{1}\mathbb{1}^\top - \frac{d(Y^*)}{n}\mathbb{1}\mathbb{1}^\top + \frac{d(Y^*)}{n}\mathbb{1}\mathbb{1}^\top - Y^*, 2w\mathbb{1}^\top - ww^\top\rangle
$$
$$
= \langle Y-\frac{d(Y)}{n}\mathbb{1}\mathbb{1}^\top, 2w\mathbb{1}^\top - ww^\top\rangle + \langle \frac{d(Y^*)}{n}\mathbb{1}\mathbb{1}^\top - Y^*, 2w\mathbb{1}^\top - ww^\top\rangle
$$
$$
+ \langle \frac{d(Y)}{n}\mathbb{1}\mathbb{1}^\top - \frac{d(Y^*)}{n}\mathbb{1}\mathbb{1}^\top, 2w\mathbb{1}^\top - ww^\top\rangle .
$$

Notice that

$$
\langle \frac{d(Y)}{n}\mathbb{1}\mathbb{1}^\top - \frac{d(Y^*)}{n}\mathbb{1}\mathbb{1}^\top, 2w\mathbb{1}^\top - ww^\top\rangle = n\Big(2\frac{\langle w,\mathbb{1}\rangle}{n} - \Big(\frac{\langle w,\mathbb{1}\rangle}{n}\Big)^2\Big)\Big(d(Y)-d(Y^*)\Big).
$$

By re-arranging terms, we can get

$$
\mathcal{A}\;\Big|\frac{Y,z}{8}\; n\Big(1-\frac{\langle w,\mathbb{1}\rangle}{n}\Big)^2\Big(d(Y)-d(Y^*)\Big) = \langle Y-\frac{d(Y)}{n}\mathbb{1}\mathbb{1}^\top, 2w\mathbb{1}^\top - ww^\top\rangle + \langle \frac{d(Y^*)}{n}\mathbb{1}\mathbb{1}^\top - Y^*, 2w\mathbb{1}^\top - ww^\top\rangle
$$

Squaring both sides, we get

$$
\mathcal{A}\;\Big|\frac{Y,z}{8}\; n^2\Big(1-\frac{\langle w,\mathbb{1}\rangle}{n}\Big)^4\Big(d(Y)-d(Y^*)\Big)^2
$$
$$
= \Big(\langle Y-\frac{d(Y)}{n}\mathbb{1}\mathbb{1}^\top, 2w\mathbb{1}^\top - ww^\top\rangle + \langle \frac{d(Y^*)}{n}\mathbb{1}\mathbb{1}^\top - Y^*, 2w\mathbb{1}^\top - ww^\top\rangle\Big)^2
$$
$$
\leqslant 2\langle Y-\frac{d(Y)}{n}\mathbb{1}\mathbb{1}^\top, 2w\mathbb{1}^\top - ww^\top\rangle^2 + 2\langle \frac{d(Y^*)}{n}\mathbb{1}\mathbb{1}^\top - Y^*, 2w\mathbb{1}^\top - ww^\top\rangle^2
$$
$$
\tag{C.14}
$$

By definition of $w$, it follows that $\mathcal{A}\;\big|\frac{z}{2}\; w\odot w = w$ and

$$
\mathcal{A}\;\Big|\frac{z}{2}\;\langle \mathbb{1}, w\rangle = \langle \mathbb{1}, \mathbb{1} - (\mathbb{1}-z)\odot(\mathbb{1}-z^*)\rangle
$$
$$
= n - \sum_{i\in[n]}(1-z_i)(1-z_i^*)
$$
$$
= \sum_{i\in[n]} z_i + \sum_{i\in[n]} z_i^* - \sum_{i\in[n]} z_i z_i^*
$$
$$
\tag{C.15}
$$
$$
\leqslant 2\eta n .
$$

Therefore, $w$ satisfies $\mathcal{A}_{\text{label}}(w; 2\eta)$, and, by the SOS certificate in $\mathcal{A}_{\text{degree}}$, it follows that

$$\mathcal{A} \left|\frac{Y,z}{8} \left\langle Y - \frac{d(Y)}{n}\mathbb{1}\mathbb{1}^\top, 2w\mathbb{1}^\top - ww^\top\right\rangle^2 \leqslant C_1(\eta) \cdot n^2 \cdot d(Y) + C_2(\eta) \cdot n^2\,. \tag{C.16}$$

and by Lemma C.3,

$$\mathcal{A} \left|\frac{Y,z}{8} \left\langle \frac{d(Y^*)}{n}\mathbb{1}\mathbb{1}^\top - Y^*, 2w\mathbb{1}^\top - ww^\top\right\rangle^2 \leqslant C_1(\eta) \cdot n^2 \cdot d(Y^*) + C_2(\eta) \cdot n^2\,. \tag{C.17}$$

Plugging Eq. (C.16) and Eq. (C.17) into Eq. (C.14), we get

$$\mathcal{A} \left|\frac{Y,z}{8} \, n^2\left(1 - \frac{\langle w, \mathbb{1}\rangle}{n}\right)^4 \left(d(Y) - d(Y^*)\right)^2 \leqslant C_1(\eta) \cdot n^2 \cdot d(Y) + C_1(\eta) \cdot n^2 \cdot d(Y^*) + 2C_2(\eta) \cdot n^2\,.$$

Dividing both sides by $n^2$, we get

$$\mathcal{A} \left|\frac{Y,z}{8} \left(1 - \frac{\langle w, \mathbb{1}\rangle}{n}\right)^4 \left(d(Y) - d(Y^*)\right)^2 \leqslant C_1(\eta) \cdot d(Y) + C_1(\eta) \cdot d(Y^*) + 2C_2(\eta)\,.$$

Plugging in $\mathcal{A} \left|\frac{z}{2} \langle \mathbb{1}, w\rangle \leqslant 2\eta n$ from Eq. (C.15), we get

$$\mathcal{A} \left|\frac{Y,z}{8} (1 - 2\eta)^4 \left(d(Y) - d(Y^*)\right)^2 \leqslant C_1(\eta) \cdot d(Y) + C_1(\eta) \cdot d(Y^*) + 2C_2(\eta)\,. \tag{C.18}$$

Notice that, the following holds for any $C_\eta > 0$,

$$d(Y) = \frac{d(Y) - d(Y^*)}{\sqrt{C_\eta \cdot d(Y^*)}} \cdot \sqrt{C_\eta \cdot d(Y^*)} + d(Y^*)$$

$$\leqslant \frac{1}{2}\left(\frac{\left(d(Y) - d(Y^*)\right)^2}{C_\eta \cdot d(Y^*)} + C_\eta \cdot d(Y^*)\right) + d(Y^*)$$

$$= \frac{\left(d(Y) - d(Y^*)\right)^2}{2C_\eta \cdot d(Y^*)} + \left(\frac{C_\eta}{2} + 1\right)d(Y^*)$$

$$\leqslant \frac{\left(d(Y) - d(Y^*)\right)^2}{2C_\eta C_{Y^*}} + \left(\frac{C_\eta}{2} + 1\right)d(Y^*)\,,$$

Plugging this into Eq. (C.18), it follows that

$$\mathcal{A} \left|\frac{Y,z}{8} (1-2\eta)^4 \left(d(Y)-d(Y^*)\right)^2 \leqslant C_1(\eta)\frac{\left(d(Y) - d(Y^*)\right)^2}{2C_\eta C_{Y^*}} + C_1(\eta)\left(\frac{C_\eta}{2} + 2\right)d(Y^*) + 2C_2(\eta)\,. \tag{C.19}$$

Let $C_\eta = \frac{50 C_1(\eta)}{C_{Y^*}(1-2\eta)^4}$. Since $\eta^2 \log(1/\eta) < 1$ for $\eta \in [0, \frac{1}{2})$, we have $C_1(\eta) = C_{\text{deg}}\eta^2 \log(1/\eta) < C_{\text{deg}}$ and $C_\eta < \frac{50 C_{\text{deg}}}{C_{Y^*}(1-2\eta)^4}$. Plugging these into Eq. (C.19), it follows that

$$\mathcal{A} \left|\frac{Y,z}{8} (1 - 2\eta)^4 \left(d(Y) - d(Y^*)\right)^2$$

$$\leqslant \frac{(1-2\eta)^4}{100}\left(d(Y) - d(Y^*)\right)^2 + C_1(\eta)\left(\frac{25 C_{\text{deg}}}{C_{Y^*}(1-2\eta)^4} + 2\right)d(Y^*) + 2C_2(\eta)$$

$$\leqslant \frac{(1-2\eta)^4}{100}\left(d(Y) - d(Y^*)\right)^2 + \frac{C_1(\eta)(25 C_{\text{deg}}/C_{Y^*} + 2)}{(1-2\eta)^4}d(Y^*) + 2C_2(\eta)\,.$$

Rearranging the terms, it follows that

$$\mathcal{A} \left|\frac{Y,z}{8} \, 0.99(1 - 2\eta)^4 \left(d(Y) - d(Y^*)\right)^2 \leqslant \frac{C_1(\eta)(25 C_{\text{deg}}/C_{Y^*} + 2)}{(1-2\eta)^4}d(Y^*) + 2C_2(\eta)$$

$$\left(d(Y) - d(Y^*)\right)^2 \leqslant \frac{C_1(\eta)(50C_{\deg}/C_{Y^*} + 4)}{(1 - 2\eta)^8} d(Y^*) + \frac{4C_2(\eta)}{(1 - 2\eta)^4}$$

$$\left(d(Y) - d(Y^*)\right)^2 \lesssim \frac{C_1(\eta)}{(1 - 2\eta)^8} d(Y^*) + \frac{C_2(\eta)}{(1 - 2\eta)^4}.$$

Plugging in $C_1(\eta) = C_{\deg}\eta^2 \log(1/\eta)$ and $C_2(\eta) = C_{\deg}\eta^2 \log^2(1/\eta)$, it follows that

$$\mathcal{A}\big|_{8}^{Y,z} \left(d(Y) - d(Y^*)\right)^2 \lesssim \frac{C_{\deg}\eta^2 \log(1/\eta)}{(1 - 2\eta)^8} d(Y^*) + \frac{C_{\deg}\eta^2 \log^2(1/\eta)}{(1 - 2\eta)^4}$$

$$\lesssim \frac{\eta^2 \log(1/\eta)}{(1 - 2\eta)^8} d(Y^*) + \frac{\eta^2 \log^2(1/\eta)}{(1 - 2\eta)^4}.$$

$\square$

## D   Robust binomial mean estimation

In this section, we show that the sample median can robustly estimate the mean of a binomial distribution. For simplicity, we prove the result for a smaller but rich enough parameter regime than our main theorem Theorem 1.2. We also make basic integrality assumptions to avoid having to deal with floors and ceilings throughout the proof. To compare it with Theorem 1.2, we set $m = n$ in the arguments below.

The corrupted binomial model that we consider is defined as follows.

**Definition D.1** ($\eta$-corrupted $(m, n, d)$-binomial model)**.**  Let $\eta \in [0, 1]$, the $\eta$-corrupted $(m, n, d)$-binomial model is generated by first sampling $m$ i.i.d samples $X_1^\circ, X_2^\circ, \ldots, X_m^\circ$ from $\text{Bin}(n, \frac{d}{n})$, then adversarially picking an $\eta$-fraction of the samples and arbitrarily modifying them to get $X_1, X_2, \ldots, X_m$.

The goal of robust binomial mean estimation is to estimate the mean $d$ given observation of corrupted samples $X_1, X_2, \ldots, X_m$ that are generated according to the $\eta$-corrupted $(m, n, d)$-binomial model from Definition D.1. We will show that the median of the corrupted samples satisfy the following error guarantee.

**Theorem D.2.**  *Given $\eta$-corrupted $(m, n, d)$-binomial samples $X_1, X_2, \ldots, X_m$, when $10000 \leqslant d \leqslant 0.0001n$ is an integer and $\frac{1000}{\sqrt{m}} \leqslant \eta \leqslant 0.01$, the median $\hat{d}$ satisfies, with probability $1 - 4\exp(-\eta^2 m/4)$,*

$$\left|\hat{d} - d\right| \leqslant O(\eta\sqrt{d}).$$

To prove Theorem D.2, we need the following two lemmas.

**Lemma D.3.**  *When $10000 \leqslant d \leqslant 0.0001n$ is an integer and $\eta \leqslant 0.01$, let $X \sim \text{Bin}(n, \frac{d}{n})$ be a binomial random variable, it follows that*

$$\mathbb{P}(d - 100\eta\sqrt{d} \leqslant X \leqslant d - 1) \geqslant 4\eta, \tag{D.1}$$

*and,*

$$\mathbb{P}(d + 1 \leqslant X \leqslant d + 100\eta\sqrt{d}) \geqslant 4\eta. \tag{D.2}$$

*Proof.*  Consider an arbitrary integer $t \in [d - 100\eta\sqrt{d}, d + 100\eta\sqrt{d}]$, it follows that

$$\log(\mathbb{P}(X = t)) = \log\left(\binom{n}{t}\left(\frac{d}{n}\right)^t \left(\frac{n - d}{n}\right)^{n-t}\right)$$

$$= \log\left(\frac{n!}{t!(n - t)!}\left(\frac{d}{n}\right)^t \left(\frac{n - d}{n}\right)^{n-t}\right) \tag{D.3}$$

$$= \log\left(\frac{n!}{t!(n - t)!}\right) + t\log\left(\frac{d}{n}\right) + (n - t)\log\left(\frac{n - d}{n}\right).$$

By Stirling's approximation, it follows that

$$
\frac{n!}{t!(n-t)!} = \frac{\sqrt{2\pi n}(n/e)^n(1+O(1/n))}{\sqrt{2\pi t}(t/e)^t(1+O(1/t)) \cdot \sqrt{2\pi(n-t)}((n-t)/e)^{n-t}(1+O(1/(n-t)))}
$$

$$
= \sqrt{\frac{n}{2\pi t(n-t)}} \cdot \frac{1+O(1/n)}{(1+O(1/t))(1+O(1/(n-t)))} \cdot \left(\frac{n}{t}\right)^t \left(\frac{n}{n-t}\right)^{n-t} \qquad \text{(D.4)}
$$

$$
\geqslant \sqrt{\frac{1}{10t}}\left(\frac{n}{t}\right)^t\left(\frac{n}{n-t}\right)^{n-t}.
$$

Plugging Eq. (D.4) into Eq. (D.3), we get

$$
\log(\mathbb{P}(X=t)) \geqslant \frac{1}{2}\log\left(\frac{1}{10t}\right) + t\log\left(\frac{d}{t}\right) + (n-t)\log\left(\frac{n-d}{n-t}\right). \qquad \text{(D.5)}
$$

For the second term $t\log\left(\frac{d}{t}\right)$, we use the Maclaurin series of natural logarithm and get

$$
t\log\left(\frac{d}{t}\right) = t\log\left(1+\frac{d-t}{t}\right)
$$

$$
\geqslant t\left(\frac{d-t}{t} - \frac{(d-t)^2}{2t^2} + \frac{(d-t)^3}{3t^3}\right)
$$

$$
= d - t - \frac{(d-t)^2}{2t} + \frac{(d-t)^3}{3t^2}.
$$

Since $(d-t)^2 \leqslant 10000\eta^2 d \leqslant d$ and $t \geqslant d - 100\eta\sqrt{d} \geqslant 0.99d$, it follows that

$$
t\log\left(\frac{d}{t}\right) \geqslant d - t - 0.9. \qquad \text{(D.6)}
$$

For the last term $(n-t)\log\left(\frac{n-d}{n-t}\right)$, we can also use the Maclaurin series of natural logarithm and get

$$
(n-t)\log\left(\frac{n-d}{n-t}\right) = (n-t)\log\left(1+\frac{t-d}{n-t}\right)
$$

$$
\geqslant (n-t)\left(\frac{t-d}{n-t} - \frac{(t-d)^2}{2(n-t)^2} + \frac{(t-d)^3}{3(n-t)^3}\right)
$$

$$
= t - d - \frac{(t-d)^2}{2(n-t)} + \frac{(t-d)^3}{3(n-t)^2}.
$$

Since $(t-d)^2 \leqslant 10000\eta^2 d \leqslant 0.0001n$, it follows that

$$
(n-t)\log\left(\frac{n-d}{n-t}\right) \geqslant t - d - 0.1. \qquad \text{(D.7)}
$$

Plugging Eq. (D.6) and Eq. (D.7) into Eq. (D.5), we get

$$
\log(\mathbb{P}(X=t)) \geqslant \frac{1}{2}\log\left(\frac{1}{10t}\right) - 1 \geqslant \frac{1}{2}\log\left(\frac{1}{t}\right) - 3. \qquad \text{(D.8)}
$$

Now, we are ready to prove Eq. (D.1) and Eq. (D.2). We first consider the regime $t \in [d - 100\eta\sqrt{d}, d-1]$, it follows that

$$
\log\left(\mathbb{P}(d - 100\eta\sqrt{d} \leqslant X \leqslant d - 1)\right) = \log\left(\sum_{t=d-100\eta\sqrt{d}}^{d-1} \mathbb{P}(X=t)\right)
$$

$$
\geqslant \log\left(100\eta\sqrt{d}\,\mathbb{P}(X = d - 100\eta\sqrt{d})\right)
$$

$$= \log(100\eta) + \frac{1}{2}\log(d) + \log\left(\mathbb{P}(X = d - 100\eta\sqrt{d})\right)$$

$$\geqslant \log(100\eta) + \frac{1}{2}\log(d) + \frac{1}{2}\log\left(\frac{1}{d - 100\eta\sqrt{d}}\right) - 3$$

$$= \log(4\eta) + \frac{1}{2}\log\left(\frac{d}{d - 100\eta\sqrt{d}}\right)$$

$$\geqslant \log(4\eta),$$

which implies that

$$\mathbb{P}(d - 100\eta\sqrt{d} \leqslant X \leqslant d - 1) \geqslant 4\eta.$$

The regime $t \in [d + 1, d + 100\eta\sqrt{d}]$ can be proved in a similar way

$$\log\left(\mathbb{P}(d + 1 \leqslant X \leqslant d + 100\eta\sqrt{d})\right) = \log\left(\sum_{t=d+1}^{d+100\eta\sqrt{d}} \mathbb{P}(X = t)\right)$$

$$\geqslant \log\left(100\eta\sqrt{d}\,\mathbb{P}(X = d + 100\eta\sqrt{d})\right)$$

$$= \log(100\eta) + \frac{1}{2}\log(d) + \log\left(\mathbb{P}(X = d + 100\eta\sqrt{d})\right)$$

$$\geqslant \log(100\eta) + \frac{1}{2}\log(d) + \frac{1}{2}\log\left(\frac{1}{d + 100\eta\sqrt{d}}\right) - 3$$

$$= \log(4.1\eta) + \frac{1}{2}\log\left(\frac{d}{d + 100\eta\sqrt{d}}\right)$$

$$\geqslant \log(4.1\eta) + \frac{1}{2}\log\left(\frac{1}{1.01}\right)$$

$$\geqslant \log(4\eta).$$

which implies that

$$\mathbb{P}(d + 1 \leqslant X \leqslant d + 100\eta\sqrt{d}) \geqslant 4\eta.$$

$\square$

**Lemma D.4.** *When $10000 \leqslant d \leqslant 0.0001n$ is an integer and $\frac{1000}{m} \leqslant \eta \leqslant 0.01$, let $X_1^\circ, X_2^\circ, \ldots, X_m^\circ$ be $m$ i.i.d. binomial random variables from $\mathrm{Bin}(n, \frac{d}{n})$, with probability $1 - 2\exp\left(-\frac{\eta m}{2}\right)$, there are at least $2\eta m$ samples in range $[d - 100\eta\sqrt{d}, d - 1]$ and at least $2\eta m$ samples in range $[d + 1, d + 100\eta\sqrt{d}]$.*

*Proof.* By Lemma D.3, we know that for each $X \in \{X_1^\circ, X_2^\circ, \ldots, X_m^\circ\}$, we have

$$\mathbb{P}(d - 100\eta\sqrt{d} \leqslant X \leqslant d - 1) \geqslant 4\eta, \tag{D.9}$$

and,

$$\mathbb{P}(d + 1 \leqslant X \leqslant d + 100\eta\sqrt{d}) \geqslant 4\eta. \tag{D.10}$$

Let us denote by $Z_i$ the event that $X_i^\circ$ is in range $[d - 100\eta\sqrt{d}, d - 1]$. By Eq. (D.9), we have $\mathbb{E}[Z_i] \geqslant 4\eta$. Since $X_1^\circ, X_2^\circ, \ldots, X_m^\circ$ are i.i.d., the events $Z_i$'s are i.i.d. Bernoulli random variables. Therefore, by Chernoff bound, it follows that

$$\mathbb{P}\left(\sum_i Z_i \leqslant 2\eta m\right) \leqslant \mathbb{P}\left(\sum_i Z_i \leqslant (1 - \frac{1}{2})\mathbb{E}\left[\sum_i Z_i\right]\right)$$

$$\leqslant \exp\left(-\frac{\mathbb{E}[\sum_i Z_i]}{8}\right)$$

$$\leqslant \exp\left(-\frac{\eta m}{2}\right).$$

Therefore, with probability at least $1 - \exp\left(-\frac{\eta m}{2}\right)$, at least $2\eta m$ samples are in range $[d - 100\eta\sqrt{d}, d - 1]$. Using the same argument, it can also be shown that, with probability at least $1 - \exp\left(-\frac{\eta m}{2}\right)$, at least $2\eta m$ samples are in range $[d + 1, d + 100\eta\sqrt{d}]$. The lemma follows by union bound. $\square$

Now, we are ready to prove Theorem D.2 for the error guarantee of median for corrupted binomial samples.

*Proof of Theorem D.2.* Consider the uncorrupted samples $X_1^\circ, X_2^\circ, \ldots, X_m^\circ$. Notice that the median of $\mathrm{Bin}(n, \frac{d}{n})$ is $d$ when $d$ is an integer, that is $\mathbb{P}(X_i^\circ \geqslant d) \geqslant \frac{1}{2}$ and $\mathbb{P}(X_i^\circ \leqslant d) \geqslant \frac{1}{2}$ for each $i \in [m]$. Using similar arguments via Chernoff bound as Lemma D.4, it is easy to check that with probability at least $1 - 2\exp(-\eta^2 m/4)$, there are at least $\frac{(1-\eta)m}{2}$ uncorrupted samples that are at least $d$ and at least $\frac{(1-\eta)m}{2}$ uncorrupted samples that are at most $d$.

By Lemma D.4, with probability $1 - 2\exp(-\eta m/2) \geqslant 1 - 2\exp(-\eta^2 m/4)$, there are at least $2\eta m$ uncorrupted samples in range $[d - 100\eta\sqrt{d}, d - 1]$ and at least $2\eta m$ uncorrupted samples in range $[d + 1, d + 100\eta\sqrt{d}]$.

Therefore, combining the two bounds, with probability $1 - 4\exp(-\eta^2 m/4)$, there are at least $\frac{m}{2} + \frac{3\eta m}{2}$ uncorrupted samples in range $[d - 100\eta\sqrt{d}, n]$ and at least $\frac{m}{2} + \frac{3\eta m}{2}$ uncorrupted samples in range $[0, d + 100\eta\sqrt{d}]$.

After corrupting $\eta m$ samples, there are still at least $\frac{m}{2} + \frac{\eta m}{2}$ samples in range $[d - 100\eta\sqrt{d}, n]$ and at least $\frac{m}{2} + \frac{\eta m}{2}$ samples in range $[0, d + 100\eta\sqrt{d}]$. Thus, the median $\hat{d}$ satisfies, with probability $1 - 4\exp(-\eta^2 m/4)$,

$$\left|\hat{d} - d\right| \leqslant O(\eta\sqrt{d}).$$

$\square$

