# OpenReview forum: "Improved Robust Estimation for Erdős-Rényi Graphs: The Sparse Regime and Optimal Breakdown Point"
_NeurIPS.cc/2025/Conference — NeurIPS 2025 poster_

### Official Review · Reviewer_mXLy · 2025-07-01

**Clarity:** 3
**Significance:** 4
**Originality:** 4
**Rating:** 3
**Confidence:** 4

**Summary:**

This paper addresses the problem of robustly estimating the edge density parameter $d^\circ$ of Erdős-Rényi random graphs $G(n, d^\circ/n)$ under adversarial node corruptions. ​ The authors propose the first polynomial-time algorithm that achieves near-optimal error rates and reaches the optimal breakdown point of $ \eta = 1/2 $. ​ The algorithm leverages the sum-of-squares (SoS) hierarchy to construct certificates for concentration properties of edges in sparse graphs, overcoming limitations of previous methods in the sparse regime ($d^\circ = o(\log n)$). ​ The paper provides formal proofs, theoretical guarantees, and a clean analysis of the algorithm's performance.

**Questions:**

Below are my detailed comments/questions:

1. For the sake of completeness, it will be very useful for the authors to explain why robust mean estimation methods will not work in this setting. The paper has referred to dependencies of the degrees of the uncorrupted graph. While that is true, I cannot understand the limitation of the following simple trick: If I view the set of edges as a (n choose 2) binary vector, then each entry of this vector is iid generated with prob. $d^\circ/n$. Therefore, estimating the mean is a simple 1D mean estimation problem - if some elements are corrupted, then why will median of means not work? median of means is optimal for robust mean estimation in 1D.

2. While the results are strong, the lack of experiments is a glaring problem. Given that the setting is simple, the lack of experiments (even synthetic) hurts the value of the paper. Especially so since the authors have highlighted the efficiency and statistical guarantees of existing work as a limitation.

**Ethical Concerns:**

["NO or VERY MINOR ethics concerns only"]

**Final Justification:**

There are two main reasons

1. The limitation of an existing statistically optimal approach has not been discussed in the context of this estimation problem
2. The lack of experiments is glaring espescially since the authors have touted efficiency to be one of the main advantages

**Limitations:**

Simple Experiments even on synthetic datasets

**Quality:**

3

**Strengths And Weaknesses:**

The paper is well-written and well-motivated. However, the lack of any sort of experiments (even synthetic) is glaring

---

> ### Author Rebuttal · Authors · 2025-07-29
>
> Thank you for your review and positive comments!
>
> Regarding the relation to 1-D robust mean estimation: On a very high level the issue is that the parameter we want to estimate is $n$ times the mean of the Bernoulli distribution.
>
> One way to see that this strategy will incur very sub-optimal errors is as follows. If we do not consider any structural properties of the corruptions and treat the problem as that of robustly estimating the parameter of a Bernoulli distribution, this can at best get error $\eta n$. Even in the easier corruption model in which each sample is drawn from $Ber(\tfrac d  n)$ with probability $1-\eta$ and some error distribution with probability $\eta$, it is not possible to get error better than $\eta n$ (note that the adversary can simulate this model up to constant shifts in $\eta$). In particular, we can choose the error distribution to be $Ber(1)$, then the resulting input distribution is roughly $Ber(\tfrac d n + \eta) = Ber(\tfrac {\hat{d}} n)$ for $\hat{d} \approx \eta n$. Thus, $\lvert d - \hat{d} \rvert \geq \eta n$ but we cannot tell the two models apart.
>
> For more intuition, consider the problem of robustly estimating the mean of a 1-D Gaussian with standard deviation $\sigma$, given an $\eta$-corrupted sample. Then, the optimal error we can achieve is $\Omega(\sigma \cdot \eta)$. I.e., there are two Gaussian distributions $G_1, G_2$ with mean $\mu_1, \mu_2$ and standard deviations $\sigma_1, \sigma_2$, such that $\lvert \mu_1 - \mu_2 \rvert \geq \Omega(\min(\sigma_1, \sigma_2) \cdot \eta)$, but we cannot distinguish $G_1$ and $G_2$. We expect a similar lower bound for the i.i.d. Bernoulli case. In this case, the means would be $d_1/ n$ and $d_2/n$ and we get that $\lvert \tfrac {d_1} n - \tfrac {d_2} n \rvert \geq \Omega \left ( \sqrt{\tfrac{\min (d_1, d_2) } n}\eta \right )$. This implies that $\lvert d_1 - d_2 \rvert \geq  \Omega \left ( \sqrt{\min ( d_1, d_2 ) }  \sqrt{n}\cdot \eta \right )$ which is much larger than the error we obtain.

---

> > ### Comment · Reviewer_mXLy · 2025-08-05
> > **Follow-up**
> >
> > The authors are pointing out to a limitation of existing approaches in robust mean estimation even in 1D - to my understanding, this limitation needs to be understood in a much better way. The authors have not cited any papers which showcase the above limitation. In that sense, I could not go through the math to understand the limitation carefully.
> >
> > As before, lack of experiments is glaring. I expected authors to provide some kind of synthetic experiments.

---

> > > ### Author Response · Authors · 2025-08-05
> > >
> > > Thanks for your feedback.
> > >
> > > Regarding references for the 1-D discussion: The claim that the Huber contamination model can be simulated by our adversarial model can be found in Claim 2.5 in "Robust Estimators in High Dimensions without the Computational Intractability" by Diakonikolas, Kamath, Kane, Li, Moitra, Stewart. A proof of the 1-D Gaussian lower bound can be found in Lemma 1.9 in "Algorithmic High-Dimensional Robust Statistics" by Diakonikolas and Kane. We will add a clear discussion about this to the paper, thank you for pointing this out.
> > >
> > > Regarding experiments: Our approach achieves (nearly) optimal guarantees in polynomial time (in $n$). Previous algorithms achieving such guarantees needed exponential time. In this sense, our algorithm is much more time-efficient than previous works. More specifically, the runtime is a large polynomial in the input size. Degree-$\ell$ sum-of-squares programs can be solved in time $n^{O(\ell)}$, where the constant in the big $O(\cdot)$ is quite large (and comes from solving a semi-definite program with $n^{\ell}$ variables), for us $\ell = 8$. Since semi-definite program solvers are notoriously slow, this is a bottleneck to running experiments.
> > >
> > > We do agree that finding a more practical algorithm achieving similar guarantees is a very good follow-up question. Indeed, in the past, e.g., for robust Gaussian mean estimation, there were fast iterative algorithms that used the same principles as the SoS proofs-to-algorithms paradigm (see the book "Algorithmic High-Dimensional Robust Statistics" by Diakonikolas and Kane). These usually have runtime that is at most quadratic or cubic in the input size and some of these have been implemented (see, e.g., Section 7 of "Quantum Entropy Scoring for Fast Robust Mean Estimation and Improved Outlier Detection" by Dong, Hopkins, and Li and their associated GitHub repository for the code). We believe our identifiability proofs can also lead to such fast algorithms.

---

### Official Review · Reviewer_uhWe · 2025-07-02

**Clarity:** 3
**Significance:** 3
**Originality:** 3
**Rating:** 5
**Confidence:** 3

**Summary:**

This paper studies a robust statistics problem on Erdos-Renyi random graphs.  Suppose that we draw an ER graph with expected degree $d$, but then we allow an adversary to choose an arbitrary subset of $\eta n$ nodes and completely change the edges incident on them (not just between them).  Can we still estimate the original parameter $d$?  This paper gives state of the art results, showing a polytime algorithm that essentially matches the information-theoretic lower bound (up to a log factor) for all possible regimes.  This is in contrast to previous work, which either gave optimal bounds for only some parameter regimes or gave suboptimal bounds.  Their algorithm is based on a sum-of-squares proof, where they first given an inefficient algorithm, but then identify from this algorithm a certificate that can be expressed as sum-of-squares.  Since there is a well-known duality between SoS proofs and certain semidefinite programs, this means that there is an SDP-based polytime algorithm for the problem.

**Questions:**

- Is this algorithm practical?
- What's the motivation for this precise problem?

**Ethical Concerns:**

["NO or VERY MINOR ethics concerns only"]

**Final Justification:**

I gave this paper a 5 before rebuttal, and I still feel like this paper should be accepted.

**Limitations:**

yes

**Quality:**

4

**Strengths And Weaknesses:**

Strengths:
- This paper seems to basically close this problem (up to the log factor), as they get almost optimal bounds in all regimes where bounds are possible.
- The paper is well-written and understandable.

Weaknesses:
- While I understand that this is a theory paper, for NeurIPS I would expect there to be either experiments or some type of argument about practicality.  But my understanding is that SDP solvers are still quite slow, so the algorithm in this paper might not actually be practical.
- There has been previous work on this problem, so it seems to be of interest to the community, but I wish the authors had bothered to motivate the problem a little bit.  Why this corruption model?  Why study ER graphs?  Naivel, I would have expected something like "real-world graphs are often preferential-attachment or something similar, and so we focus on trying to statistically recover the parameters of those models in the presence of adversarial noise.  The exact model of adversarial noise depends on the application: one could imagine edges being corrupted, nodes being corrupted, or some combination of the two.  Here are motivating examples of why we might want to consider each of these types of corruption in each of these random graph models: ...".

---

> ### Author Rebuttal · Authors · 2025-07-29
>
> Thank you for your review and positive comments!
>
> Regarding the first weakness/question on practicality of the algorithm, the runtime of our sum-of-squares (SoS) algorithm is a large polynomial in the input size as it requires implementing large scale semidefinite programmings. Degree-$\ell$ sum-of-squares (SoS) programs can be solved in time $n^{O(\ell)}$, where the constant in the big $O(\cdot)$ is quite large (and comes from solving a semi-definite program with $n^{O(\ell)}$ variables), for us $\ell = 8$. In the past, e.g., for robust Gaussian mean estimation, there were fast iterative algorithms that are inspired by the SoS certificates from the proofs-to-algorithms paradigm. These usually have runtime that is at most quadratic or cubic in the input size. We believe our identifiability proofs can also lead to such fast algorithms.
>
> Regarding the second weakness/question on motivation for the problem, the deviation of real-world networks from the Erdős–Rényi model can often be effectively captured by node corruptions. For instance, in social networks, individuals or organizations may create fake profiles to manipulate network statistics, thereby misleading recommendation systems for their own purposes. In communication systems, malfunctioning devices can emit arbitrary or adversarial signals that interfere with accurate analysis. In biological networks, contaminated experimental samples may exhibit spurious interactions with other proteins, distorting the underlying structure. While Erdős–Rényi graphs are simplified, idealized models of real-world networks, we view them as a fundamental first step toward addressing these critical challenges in real-world settings.

---

### Official Review · Reviewer_ighP · 2025-07-03

**Clarity:** 3
**Significance:** 4
**Originality:** 4
**Rating:** 5
**Confidence:** 3

**Summary:**

The paper studies the problem of estimating the edge density of ER random graphs $G(n, d/n)$ under node corruptions. The paper proves that their method can estimate d within an additive error of $O((\sqrt{\log{(n)}/n}+\eta \sqrt{\log{(1/\eta)}})\sqrt{d}+\eta \log{(1/\eta)})$. This upper bound differs from the lower bound for this problem by a factor of $\log{(1/\eta)}$. The paper also emphasizes that the proposed algorithm works for all $d \ge \Omega(1)$ and can achieve optimal breakdown point $\eta =1/2$.

**Questions:**

I think this paper is a clear accept and have no specific questions.

**Ethical Concerns:**

["NO or VERY MINOR ethics concerns only"]

**Final Justification:**

After reading the paper and the authors' rebuttal, I decided to keep my score.

**Limitations:**

There is not much concern about the potential negative societal impact of this paper.

**Paper Formatting Concerns:**

I think the paper follows the NeurIPS 2025 Paper Formatting Instructions.

**Quality:**

4

**Strengths And Weaknesses:**

Strengths

S1. The paper studies the problem of estimating the edge density of Erdős–Rényi (ER) random graphs — a fundamental problem with both theoretical and practical significance. It improves the approximation error bound for this task. The prior bound in [AJK+] is suboptimal and does not recover the non-robust case when there are no corruptions. I think the paper is strong and is worthy of acceptance.

S2. The paper is well written. Its structure is clear and easy to follow. Prior work is thoroughly reviewed, and the high-level intuition behind the proposed method is well explained.

S3. As mentioned in the paper, the proof is relatively simple, which actually enhances the contribution of the paper


Weaknesses

W1. The additive error bound presented in the paper is not very clean and still leaves a gap compared to the optimal result. There remains room for further exploration in this direction.

W2. The paper does not include any empirical study. However, since this is a theoretical paper, this is acceptable in my view.

---

> ### Author Rebuttal · Authors · 2025-07-29
>
> Thank you for your review and positive comments!
>
> Regarding the first weakness, note that the only possible improvement are in terms of factors of $\mathrm{poly}(\log(1/\eta))$. More specifically, our error upper bound can be written as $\max \left (\sqrt{d} \cdot \sqrt{\tfrac {\log n} n}, \sqrt{d} \cdot \eta \sqrt{\log (1/\eta)}, \eta \log(1/\eta) \right )$. This matches the information theoretical lower bound of $\max  \left(  \sqrt{d} \cdot \sqrt{\tfrac {\log n} n}, \sqrt{d} \cdot \eta , \eta \right)$ up to factors of $\log(1/\eta)$ and $\sqrt{\log(1/\eta)}$.
>
> In our regime, when $d \geq 1$, a clean and only slightly looser way to state our upper bound is $\sqrt{d} \cdot \left(\sqrt{\tfrac {\log n} n}+ \eta \log (1/\eta) \right)$. The first part corresponds to the "non-robust" error and the second part to the "robust" error.

---

### Official Review · Reviewer_Kp9b · 2025-07-03

**Clarity:** 3
**Significance:** 3
**Originality:** 4
**Rating:** 5
**Confidence:** 4

**Summary:**

Consider a graph G generated from the classic Erdos-Renyi distribution G(n,p), where we think of p = d/n. So the average degree will be d. An adversary corrupts an $\eta$-fraction of the nodes, by arbitrarily changing edges incident to these "corrupted nodes". The aim is to estimate d. Previous work showed that many standard estimators fail. There has been recent work getting estimators for this regime.

The main problem is that when d = Theta(1), no previous algorithm gets the optimal bound. The first result [AJK+22] has an additive O(log n) error. The next result [CDHS24] has an additive error of d/10, regardless of d. Also, these algorithms fail when $\eta$ is larger than some small constant.

This paper gives an algorithm that avoids all of these issues, and gets (nearly) optimal error. The bounds are technical, but the main point is that it allows for $\eta < 1/2$, d to be a constant, and the errors goes to zero as d becomes larger.

The main idea is to use tools for the Sum-of-Squares (SoS) machinery that builds a semi-definite program (SDP) to solve this problem.

**Questions:**

What is the running time of the algorithm?

**Ethical Concerns:**

["NO or VERY MINOR ethics concerns only"]

**Limitations:**

Yes

**Quality:**

4

**Strengths And Weaknesses:**

The primary strength is getting an estimation result for the constant d case, with optimal threshold $\eta < 1/2$, and a multiplicative error that goes to zero as d increases.

The primary weakness is that I doubt that this is a realistic algorithm. The polynomial running time is probably quite prohibitive. So this result is of somewhat narrow interest to the people who work on this problem.

---

> ### Author Rebuttal · Authors · 2025-07-29
>
> Thank you for your review and positive comments!
>
> The runtime is a large polynomial in the input size. Degree-$\ell$ sum-of-squares (SoS) programs can be solved in time $n^{O( \ell )}$, where the constant in the big $O(\cdot)$ is quite large (and comes from solving a semi-definite program with $n^{O(\ell)}$ variables), for us $\ell = 8$.
>
> In the past, e.g., for robust Gaussian mean estimation, there were fast iterative algorithms that used the same principles as the SoS proofs-to-algorithms paradigm. These usually have runtime that is at most quadratic or cubic in the input size. We believe our identifiability proofs can also lead to such fast algorithms and believe this is an interesting follow-up question.

---

> > ### Comment · Reviewer_Kp9b · 2025-08-07
> >
> > Thanks for the clarifications!

---

### Decision · Program_Chairs · 2025-09-17

**Decision:**

Accept (poster)

**Comment:**

The paper proposes a sum-of-squares (SoS) relaxation for estimating the edge density of an Erdos-Renyi graph, where an adversary can arbitrarily add/remove edges incident to a fraction of the nodes. The main claims of going from exponential time to polynomial time, matching information-theoretic lower bounds, and achieving the optimal breakdown point, are properly defended, albeit only theoretically. Independent experimental confirmation would have been great to have as well.

Opinions were mixed but mostly positive. Thus I finally decided for acceptance. The most important concern raised by several reviewers was the lack of experiments. During the discussion, the authors agree that a level-$\ell$ SoS relaxation take $n^{O( \ell )}$ time where "the constant in the big $O(.)$ is quite large". In the appendix, the authors mention a level-8 SoS relaxation.

I still believe the contribution to be important (going from exponential time to polynomial time) but the high factor in the polynomial should be acknowledged in the abstract, introduction and over the whole manuscript (not only in the appendix) in the camera-ready version. Please also take into the account the comments from all reviewers.